# An anisotropic strategy for developing polymer electrolytes endowing lithium metal batteries with electrochemo-mechanically stable interface

Jingren Gou[1,5], Kaixuan Cui[1,5], Suqing Wang [2] ✉, Zheng Zhang[1,3] ✉, Jiale Huang[4] & Haihui Wang [1] ✉

Developing versatile solid polymer electrolytes is a reasonable approach to achieving reliable lithium metal batteries but is still challenging due to the nonuniform lithium deposition associated with the sluggish $Li^+$ kinetics and insufficient mechanical strength. Herein, the concept of developing anisotropic solid polymer electrolyte is realized via integrating polymer hosts with highly oriented polyacrylonitrile nanofibers modified by $Li_{6.4}La_3Zr_{1.4}Ta_{0.6}O_{12}$ particles. The oriented composite structure is employed to homogenize $Li^+$ flux, serving as a physical barrier to resist lithium dendrites, retarding the side reaction between the electrolyte and lithium, thus endowing a compatible interface for lithium negative electrode. Correspondingly, the Li||LiFePO$_4$ cells steadily operate over 1000 cycles, delivering durable capacity retention of 91% at 170 mA g$^{-1}$. Furthermore, numerical modeling and density functional theory are combined to clarify the multiphysics interplay between the designed solid polymer electrolyte and lithium negative electrode. This work provides a perspective for constructing interface-friendly solid polymer electrolytes at an electrochemo-mechanical level.

Stimulated by the rapid achievement in consumer electronics and intelligent energy storage systems, exploiting rechargeable batteries with high energy density has been a critical issue in modern society[1]. Lithium metal batteries (LMB) are regarded as promising solutions to the urgent energy problem due to the highest theoretical capacity (3860 mAh g$^{-1}$) and lowest redox potential (-3.04 V *vs* standard hydrogen electrode) of Li metal negative electrode[2]. Meanwhile, the occasional accidents of LMBs originating from flammable liquid electrolytes (LEs) have aroused the ever-growing attention for safety problems[3]. Catering to the acute safety demand, solid-state electrolytes (SSEs) containing inorganic SSEs (ISEs) and polymer SSEs (SPEs) have been employed in LMBs to substitute for conventional LEs, thanks to their solvent-free nature in preventing fire accidents and reliable mechanical robustness in inhibiting Li dendrites[4]. Also, designing mixed conductive layer in the interface of SSE/Li is demonstrated as an effective strategy to optimize the $Li^+$ transport in Li anode interface and then enhance the cycle life of lithium metal batteries[5–8]. Among all kinds of SSEs, SPEs display distinctive merits in the conformal interface with Li anodes and the scalability in emerging flexible electronics[9]. However, the application of SPEs is far from

[1]State Key Laboratory of Chemical Engineering and Low-carbon Technology, Department of Chemical Engineering, Tsinghua University, Beijing 100084, China. [2]School of Chemistry and Chemical Engineering, South China University of Technology, Guangzhou 510640, China. [3]State Key Laboratory of Efficient Production of Forest Resources, Beijing Key Laboratory of Lignocellulosic Chemistry, Beijing Forestry University, Beijing 100083, China. [4]School of Mechanical and Electrical Engineering, Guangzhou University, Guangzhou 510000, China. [5]These authors contributed equally: Jingren Gou, Kaixuan Cui. ✉e-mail: cesqwang@scut.edu.cn; zhengzhang0527@163.com; cehhwang@tsinghua.edu.cn

large-scale commercialization due to their unavoidable drawbacks of low ionic conductivity and inadequate mechanical strength, all of which accelerate battery failure, especially in the electrolyte-electrode interface[10].

Tremendous efforts have been dedicated to solving substantial challenges faced by SPEs. One of the reasonable approaches is to optimize the topological structure of polymer molecules, such as modulating the crosslink density of polymer matrix to mechanically prohibit Li dendrites[11], designing single ion-conductor to facilitate the migrating dynamic of Li$^+$ on the anode interface[12], and tailoring functional groups to stabilize the formation of the solid electrolyte interphase (SEI)[13]. Nevertheless, optimizing the polymer matrix cannot fundamentally address the practical problems of SPEs. On the one hand, there remains a trade-off effect between the ionic conductivity and the mechanical strength of SPEs[14]. On the other hand, such designed SPEs still suffer an undesired ionic conductivity lower than $10^{-4}$ mS cm$^{-1}$ [12], so they cannot be applied at room temperature. Apart from the attempts at a molecule level, incorporating fillers in the form of 0D nanoparticles[15], 1D nanowires[16], 2D nanosheets or network[17], and 3D framework[18] into the polymer host is also a sensible strategy to achieve SPEs with all-around performances. Among those additives, nanoparticles tend to aggregate in the bulk electrolyte because of their high specific surface, rendering blocked Li$^+$ transportation and attenuated mechanical strength[19]. By comparison, porous fillers with an integrated skeleton can avoid aggregation and serve as a mechanical barrier to resist Li dendrites[20]. Meanwhile, multifunctional supporting structures have been actualized via the combination of multidimensional additives and the integration of inorganic and organic fillers[21]. However, it appears that a convention has emerged in the design principle of composite SPEs: almost all of the functional fillers randomly distribute (like particle fillers) or exhibit disorder structures (like fibrous networks) in polymer hosts to fabricate isotropic electrolytes. As revealed in recent studies[22], the separator that features oriented fibrous structures demonstrates more uniform porous features when compared to the separator with random structures, favoring homogenized ion flux and resulting in controlled electrodeposition. To our knowledge, supporting skeletons with oriented microstructures are scarcely proposed to reinforce SPEs in the existing literature[23]. Therefore, developing anisotropic SPEs with oriented supporting structures is not only interesting but also significant. Moreover, the microstructure of the supporting framework directly governs the performance of SPEs, in turn affecting the mechanical-electrochemical condition at the SPE-Li interface. In this sense, the relationship between the interfacial stability and the orientation of the supporting structure remains unexplored due to the lack of investigations on anisotropic SPEs.

Herein, we propose an anisotropic reinforcement concept by inserting oriented PAN fiber networks into the polymer matrix that was in-situ polymerized by poly(ethylene glycol) diacrylate (PEGDA)[24], succinonitrile (SN, as plastic crystals), and lithium bis(trifluoromethanesulphonyl)imide (LiTFSI), aiming to address interfacial drawbacks that derive from the insufficient electrochemical and mechanical properties of SPEs. As revealed in previous studies, both of PAN and SN encompass C≡N polar groups which are demonstrated to contribute to enhance the electrochemical window of electrolyte[25,26]. Additionally, based on the oriented structure, Li$^+$-conductible inorganic fillers (Li$_{6.4}$La$_3$Zr$_{1.4}$Ta$_{0.6}$O$_{12}$, LLZTO) are further introduced to ameliorate the functionality of the supporting skeleton[27]. As expected, the target electrolyte combines the following advantages: 1. The ordered structure acts as guiding channels to homogenize Li$^+$ plating/stripping on the anode surface. 2. The oriented fibrous structure serves as a physical barrier with aligned baffles, efficiently suppressing Li dendrites at a mechanical level. 3. LLZTO particles reduce the reactivity of the electrolyte plasticizer (SN) with the Li anode, achieving the electrochemically stable interface. Relying on the optimized electrochemo-mechanical interface induced by the as-proposed SPE, the assembled LMBs paired with different cathodes demonstrate good cycling performance and rate capability. Besides the innovation in structure design, we also develop a multiphysics models based on reconstructed 3D structures of supporting networks to conduct theoretical calculations, elucidating the relationship between the interface stability and the supporting structure of SPEs. Significantly, the present work reveals the modulation mechanism of the orientation of the supporting structure on electrolyte performances, providing a paradigm for the development of SPEs in advanced energy storage systems.

## Results
### Structure characterizations of electrolytes
Figure 1a schematically illustrates the preparation process, which mainly involves fabricating supporting networks and synthesizing solid-state electrolytes. Firstly, electrospinning was used to manufacture the polyacrylonitrile (PAN) network, the micro-structure of which was modulated by the rotating speed of the collecting drum. Subsequently, the designed fibrous structures are employed as supporting networks for solid electrolytes (SEs), which were denoted as RSE (RPN-based SE), ASE (APN-based SE) and LASE (LAPN-based SE), respectively. A readily in-situ method based on the radical polymerization reaction was used to guarantee the close contact of the electrolyte-electrode interface. Observed from SEM images in Fig. 1b and c, the low rotating speed of the collector (800 rpm) brought about a randomly oriented PAN fibrous network (RPN). As the rotating speed increased to 4500 rpm, aligned PAN arrays were stacked on the collecting paper, and a network (APN) with the upgraded orientations was obtained. Wide-angle X-ray scattering (WAXS) and small-angle X-ray scattering (SAXS) measurements were conducted to analyze the distributing orientation of PAN fibers at a statistical level[28,29]. Supplementary Fig. 1 delivers the one-dimensional (1D) WAXS profiles of RPN and APN. It was noted that the crystalline peaks at 1.1 Å$^{-1}$ assigned to the (0 0 2) plane of the PAN polymer crystal[30], simultaneously emerging as circular-averaged peaks in two-dimensional (2D) images (Fig. 1d). For RPN, the homogeneous halo of the 2D WAXS spectrum represented the uniform intensity of the circle peak, implying randomly distributed PAN crystals. Conversely, the APN shows varying intensity of crystalline peaks in orthogonal directions, indicating uniaxially oriented PAN arrays. Figure 1e presents the 2D SAXS spectra of RPN and APN. Compared to RPN, APN displayed an elliptical pattern, corroborating its aligned crystalline in the uniaxial direction. Similar to the technical parameters for fabricating APN, LLZTO-modified APN (LAPN) was manufactured by incorporating LLZTO particles into the electrospinning precursor solution. As depicted in Fig. 1f and Supplementary Fig. 2, the LLZTO particles, ~200 nm in diameter, are interspersed within the PAN nanofiber matrix, demonstrating the successful integration of PAN and LLZTO components. In addition, EDS mapping (Supplementary Fig. 3) reveals a homogeneous distribution of the constituent elements throughout the material. Specifically, C and N are uniformly dispersed within the PAN nanofibers, while La, Zr, Ta and O are predominantly localized within the LLZTO ceramic particles. Supplementary Fig. 4 delivers 1D and 2D WAXS plots of the composite network, in which the sharp crystalline peaks exhibited good agreement with the crystal structures of cubic garnets assigned to the Ia3d space group[31]. Prior to curing process, the affinity of polymer on the lithium anode was validated via the contact angle measurement (Supplementary Fig. 5). The polymerization time was determined according to the time-dependent ohmic resistance (R$_{ohm}$) of the stainless steel (SS) symmetrical cells heated at 70 °C (Supplementary Fig. 6). During the hot curing process, the R$_{ohm}$ increased at the initial stage due to the impeded Li$^+$ diffusion, which resulted from the densification of the polymer matrix. After curing for 4 h, the R$_{ohm}$ value stabilized due to the complete polymerization of PEGDA monomers,

indicating an appropriate polymerizing time of 4 h. In the SEM image from the top view of RSE, ASE and LASE (Supplementary Fig. 7), no bubbles or defects were observed since the in-situ polymerization precursor was sufficiently absorbed and infiltrated in fibrous networks,

suggesting the formation of a continuous Li⁺ conductor. Significantly, the close contact between the as-designed SPE and Li anodes was validated by SEM images of Li/Li symmetrical cells (Supplementary Fig. 8). Figure 1g and Supplementary Fig. 9 compare the mechanical

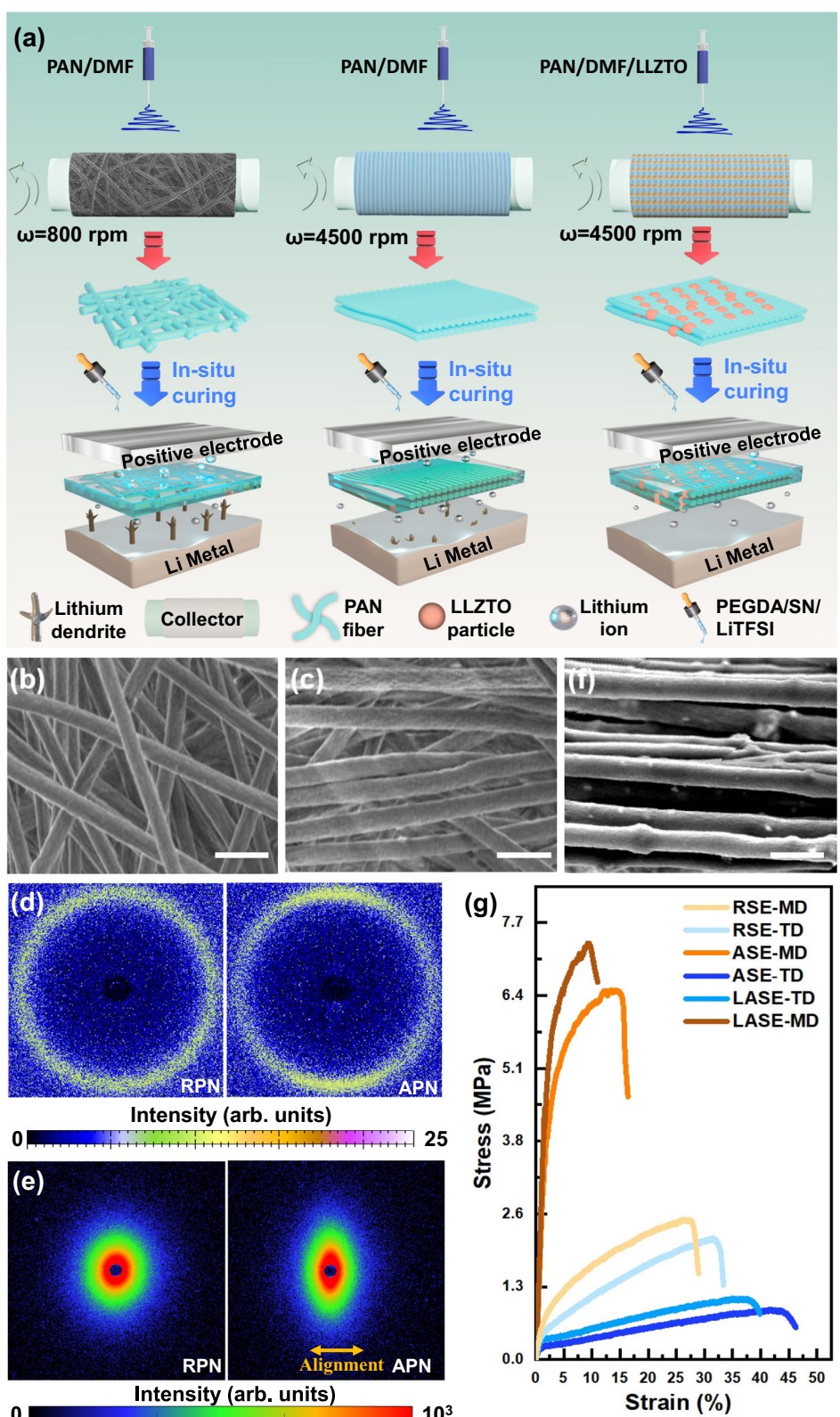

**Fig. 1 | Structure characterizations. a** Schematic diagram of the preparation procedure, which mainly involves the fabrication of supporting networks by electrospinning process and the synthesis of solid-state electrolytes by in-situ polymerization method. **b, c** SEM images of surface morphology: **b** RPN and **c** APN. Scale bars, 1 μm. **d, e** WAXS and SAXS patterns. **f** Surface morphology of LAPN. **g** Mechanical properties of the proposed electrolytes. Scale bars, 1 μm.

properties of the electrolytes. Owing to the random microstructure of RPN, RSE shows tensile strengths of 2.48 and 2.15 MPa, along with Young's moduli of 49.80 and 37.40 MPa in machine and transverse directions (MD and TD), demonstrating isotropic mechanical properties. In contrast, based on the highly oriented fibrous framework of APN, ASE exhibits anisotropic properties with tensile strengths of 6.52 and 0.85 MPa, and Young's moduli of 232.47 and 10.41 MPa at MD and TD, respectively. With the combination of LLZTO particles, the MD tensile strength and Young's modulus of LAPN further increased to 7.35 and 327.93 MPa, respectively. As schematically illustrated in Supplementary Fig. 9, the enhanced mechanical properties of the electrolyte in the uniaxial direction are conducive to practical application in large-scale production, especially in the roll-to-roll process.

### Electrochemical measurements of electrolytes

Figure 2a delivers the temperature-dependent ionic conductivity σ of the proposed electrolytes. At 25 °C, the σ values of RSE, ASE, and LASE were calculated as high as 1.06, 1.33, and 1.46 mS cm$^{-1}$, respectively, based on their EIS plots and dimension parameters (Supplementary Fig. 10 and Supplementary Table 1). The reasons for the satisfactory σ value of the electrolyte can be attributed to 1) the deep eutectic of LiTFSI and SN[32], and 2) the interaction between PEG-based polymer segments and SN, which facilitates the dissociation of lithium salts[33]. By fitting the data of σ vs. T according to the Arrhenius equation, the activation energy ($E_a$) can be obtained as 0.107, 0.094, and 0.071 eV for RSE, ASE, and LASE, respectively. In this case, a lower $E_a$ implies a lower barrier for Li$^+$ transporting through the electrolyte. To clarify the improved σ and $E_a$ of the electrolyte functionalized by LAPN, finite element simulations were implemented to model the diffusion behavior of Li$^+$ in electrolytes. The 2D computational domains were distinguished by the morphological information of the supporting networks (Supplementary Fig. 11). In the case of RSE, its cross-section can be defined as a rectangle embedded with randomly distributed ellipses due to the disordered fibrous structure of RPN. For ASE, the cross-section of the inserted network is converted into uniformly distributed circles because of the highly oriented fiber arrays. As for LASE, to describe the combination of aligned fibers and inorganic particles, the model was characterized as uniformly distributed circles with alternating sizes, the larger corresponded to the cross-section of LLZTO particles, and the smaller corresponded to the cross-section of PAN fibers. Figure 2b illustrates the computational findings, where the dashed lines indicate one of the migration pathways for Li$^+$. The simulations demonstrate that in comparison to RSE, ASE offers shorter pathways as Li$^+$ transport through the electrolyte. However, since PAN does not conduct ions, cations in ASE still need to navigate around PAN fibers during diffusion across the electrolyte. In contrast, LASE can provide additional pathways for Li$^+$ by incorporating LLZTO particles, significantly enhancing Li$^+$ diffusion efficiency and reducing the internal resistance. These simulations highlight that the oriented structure and the inclusion of LLZTO particles synergistically contribute to the higher ion conductivity and the lower diffusion barrier of SPEs.

To assess the electrochemical stability of electrolytes against Li metals, Li-Li symmetric cells were assembled to examine EIS pots as a function of storing time. As shown in Fig. 2c, the impedance of RSE and ASE displayed a more noticeable increment than that of LASE. To further quantitatively characterize the impedance variation, the EIS was fitted by equivalent circuits. Herein, $R_b$ represented the bulk resistance, reflecting the ohmic resistance of electrolytes. $R_s$ was defined as the resistance of the solid electrolyte interphase (SEI) layer (Supplementary Fig. 12). $R_{ct}$ is the charge transfer resistance, corresponding to the impedance resulting from the Li$^+$ transporting through the electrode-electrolyte interface (Supplementary Fig. 13). As a result, the interface resistance $R_{int}$ can be expressed by $R_{int} = R_s + R_{ct}$. From Fig. 2c, Li|LASE|Li delivered a small $\Delta R_{int}$ of 162.93 Ω after storing

for 21 days, in sharp contrast to RSE and ASE with drastically increased $\Delta R_{int}$ values of 440.67 and 372.51 Ω, signifying the necessity of LLZTO particles in maintaining the compatible interface between the electrolyte and Li anodes.

The electrochemical stability window of electrolytes was characterized via linear sweep voltammetry (LSV). As shown in Fig. 2d and Supplementary Fig. 14, RSE and ASE exhibited an electrochemical window of 4.70 V, much broader than PEGDA-based polymer electrolytes (4.40 V), due to the high oxidation resistance of nitrile groups belonging to PAN/SN molecules[34]. With the addition of LLZTO particles, the oxidation voltage of the inorganic-organic electrolyte further increased to 4.80 V, exhibiting potential compatibility with the high-voltage cathodes. Combining the results from chronopotentiometry and EIS measurements (Supplementary Table 2), the Li$^+$ transference number ($t_+$) of the electrolyte was calculated and the corresponding results were presented in Fig. 2e, Supplementary Fig. 15, and Supplementary Table 3. The $t_+$ of LASE was calculated as 0.55, higher than that of RSE (0.33) and ASE (0.41). According to the simulation results demonstrated in Fig. 2f and the theory of Sand's time, the electrolyte with higher $t_+$ can alleviate the Li$^+$ concentration gradient and potential gradient neighboring the anode surface, thus delaying the nucleation time of lithium dendrites and facilitating the uniform electrochemical deposition of Li[35].

### Stable anode-electrolytes interface

Galvanostatic cycling tests were conducted based on Li-Li symmetrical cells to evaluate the Li plating and stripping performances under the modulation of electrolytes. Figure 3a and b display the time-dependent polarization profiles of Li||Li cells at a current density of 0.5 mA cm$^{-2}$. At the initial stage, Li|RSE|Li exhibited a voltage plateau of about ≈64 mV. After stably cycling for 220 h, the voltage gradually increased to the maximum value of ≈73 mV at 240 h and immediately leaped to ≈9 mV as the internal short circuit occurred, which may have been caused by the uncontrolled lithium dendrites penetrating through the electrolyte. For Li|ASE|Li, the polarized potential at the initial stage was measured as ≈37 mV, exhibiting the alleviated Li reversibility compared to RSE. After a long cycle over 700 h, the cell showed fluctuating voltage profiles and broke down at 750 h. Notably, the anisotropic electrolyte based on highly oriented PAN arrays exhibited a much better ability to regulate the depositing/stripping of Li. Among the tested electrolytes, Li|LASE|Li first delivered the lowest potential of ≈35 mV and maintained the most alleviated polarization over 1500 h without severe polarization. Li|LASE|Li can still achieve a durable cycling performance over 1200 h when applied with a higher current density of 1 mA cm$^{-2}$ (in Fig. 3b). Given the above results, introducing LLZTO particles into the anisotropic electrolyte can further contribute to the stable evolution at the electrolyte-electrode interface. Figure 3c presents the surface morphology of anodes cycled with various electrolytes. Obtained from the ex-situ SEM images, the Li anode paired with RSE displayed a rugged surface with loosely porous structures, implying the uneven Li plating/stripping behavior during the cycling procedure. For the Li anode assembled with ASE, the evolved surface became relatively dense but still exhibited small burrs and fractures. Meanwhile, LASE endowed the cycled Li anode with a dense surface and without protuberant dendric structures, signifying its improved capability to modulate the propagation of Li dendrites.

To further understand the critical role of LLZTO particles in stabilizing the electrolyte-electrode interface, ex-situ X-ray photon-electron spectroscopy (XPS) measurements were performed to analyze the SEI composition of the anodes cycled with ASE and LASE for 200 h. As illustrated in Supplementary Fig. 16, the high-resolution XPS spectra of the F1s region were deconvoluted into two peaks at 688.9 and 685.3 eV, corresponding to the -CF$_3$ and LiF components of the SEI layer, respectively[36]. With increasing sputtering time, the content of LiF gradually increases. The content of LiF in the LASE electrolyte is

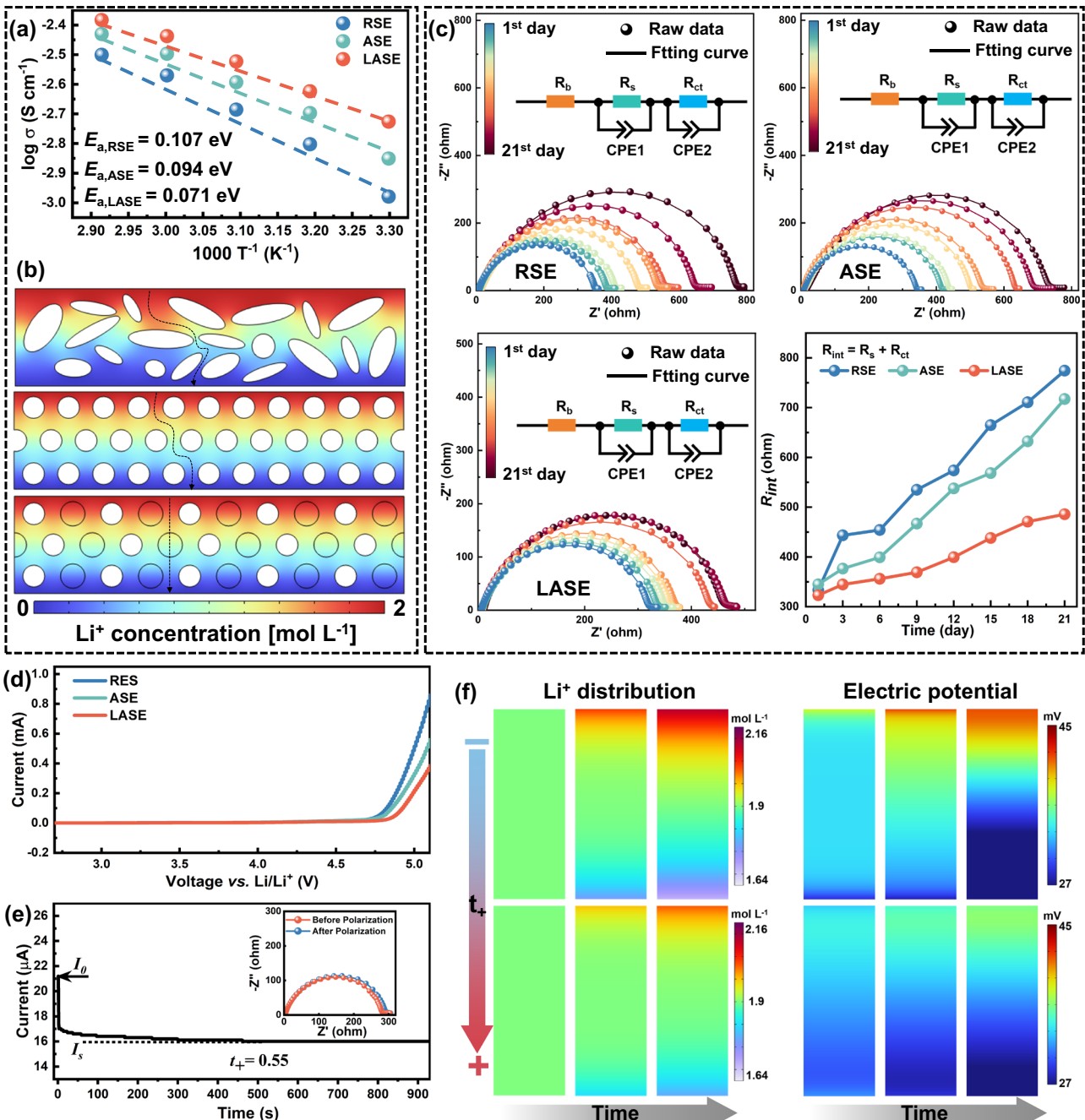

**Fig. 2 | Electrochemical performances of electrolytes. a** Temperature-dependent ionic conductivity. **b** Li+ pathway demonstrated by 2D diffusion models (RSE, ASE and LASE from up to bottom). **c** Time-dependent EIS plots and interfacial resistance (R_int). **d** Linear sweep voltammetry curves to characterize the electrochemical window. **e** Chronoamperometry and EIS results to calculate $t_+$ of LASE. **f** Numerical simulation to reveal the impact of $t_+$ on the Li+ concentration and potential distribution near the anode surface.

consistently higher than in the ASE, which helps to stabilize the interface[37]. In the high-resolution XPS spectra of the N*Is* region, the peak at 400.0 eV is attributed to the -C≡N group, resulting from the reaction between TFSI-/SN and the Li anode[38]. The peak at 399.0 eV suggests the presence of Li3N. Interestingly, a new peak at 398.0 eV, representing the -C=N-C- group, is observed, indicating a reaction between LLZTO and SN[39]. As the sputtering time increases, the content of Li3N gradually increases while the content of -C=N-C- remains relatively stable. Li3N can act as a fast Li conductor, enhancing the stability and electrical conductivity of the SEI, reducing interfacial impedance, and improving cyclic stability[40]. Further analytical

characterizations confirmed the interaction between LLZTO and SN (Supplementary Fig. 17-18). The interaction between La3+ in LLZTO and N atoms in SN can reduce the activity of -C≡N groups or convert them into weaker -C=N-C group groups, preventing the corrosion of the Li by SN molecules and enhancing the interface stability of electrolyte with Li anode[38]. The time-of-flight secondary ion mass spectrometry (ToF-SIMS) was further employed to analyze the spatial distribution of the SEI. The ToF-SIMS detected atomic fragments related to the inorganic components LiF2- and Li3N- (Fig. 3d, e and Supplementary Fig. 19). Notably, throughout the entire sputtering duration, the intensity of the inorganic component signals from the Li anode with the LASE

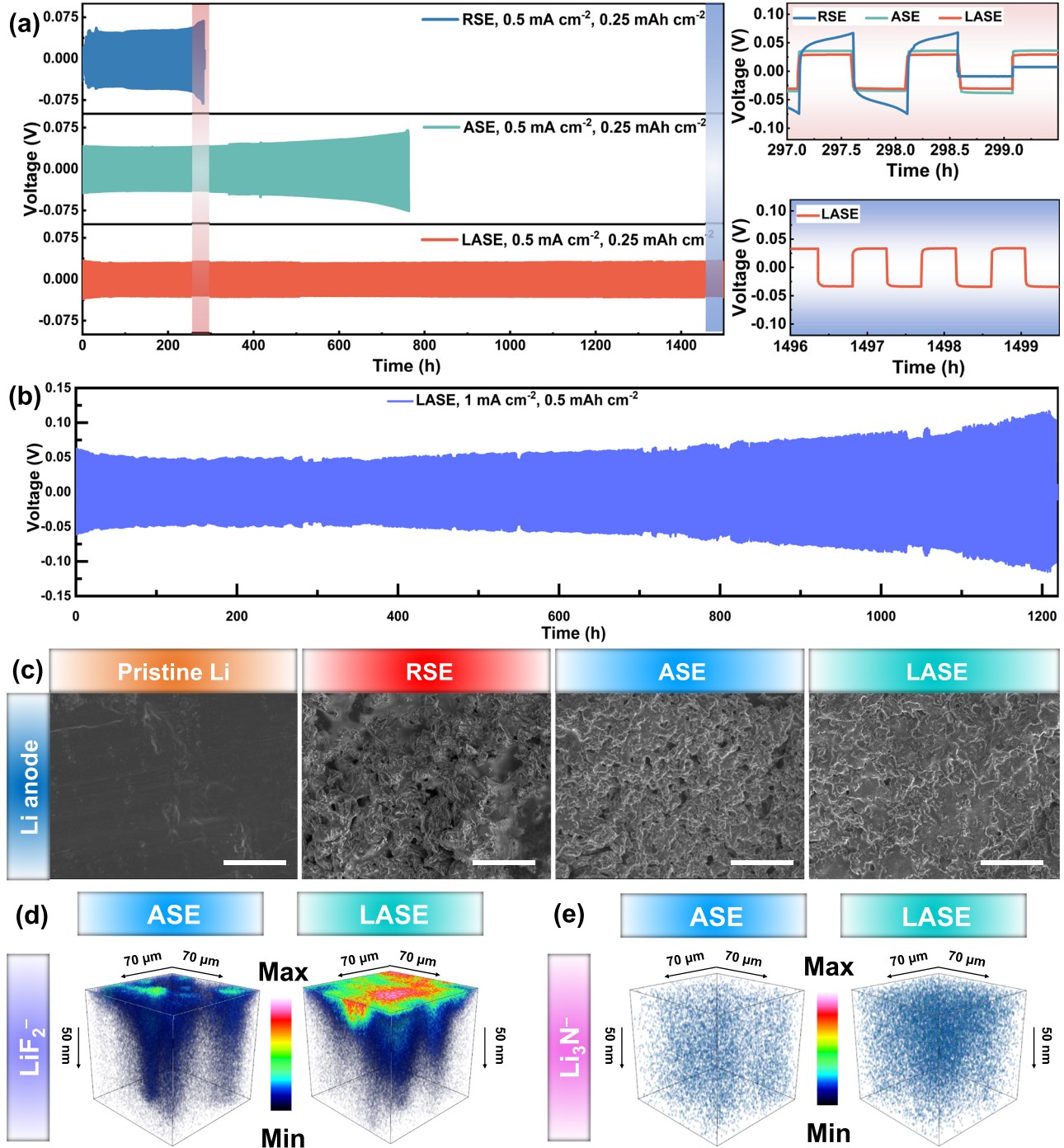

**Fig. 3 | Interfacial stability with the Li anode of electrolytes. a** Polarization profiles at the current density of 0.5 mA cm⁻². **b** Cycling performance of Li|LASE|Li cell at the current density of 1 mA cm⁻². **c** Morphology of the cycled Li anode surface cycled with various electrolytes as the capacity of 0.25 mAh cm⁻² was used. Scale bars, 5 μm. **d** ToF-SIMS 3D profiles of $LiF_2^-$ for the Li anode cycled with ASE and LASE. **e** ToF-SIMS 3D profiles of $Li_3N^-$ for the Li anode cycled with ASE and LASE.

electrolyte was significantly higher than that from the Li anode with the ASE. Moreover, the corresponding 3D profiles of the fragments indicated the formation of a uniform SEI on the Li anode, which can facilitate Li⁺ transport and suppress the growth of Li dendrites[41,42]. By comprehensively considering the aforementioned results, the comparison of RSE, ASE, and LASE is summarized in Supplementary Table 4. The anisotropic electrolyte supported by aligned PAN fibers exhibits improved uniaxial mechanical properties, ionic conductivity, and endowing ability to physically suppress Li dendrites but still preferentially react with Li metal. Furthermore, with LLZTO particles, the

anisotropic composite electrolyte displays a compatible interface with Li anodes and better voltage endurance.

## Battery performances of full cells

Full cells were assembled and measured at galvanostatic conditions to evaluate the battery performances of the as-constructed electrolytes. Figure 4a and Supplementary Fig. 20 show the cycling performances of Li ||LiFePO₄ (LFP) cells with a cathode mass loading of 3.5 mg cm⁻² at the current rate of 170 mA g⁻¹. The Li|RSE|LFP displayed rapidly plunged capacity and coulombic efficiency, finally breaking down only after

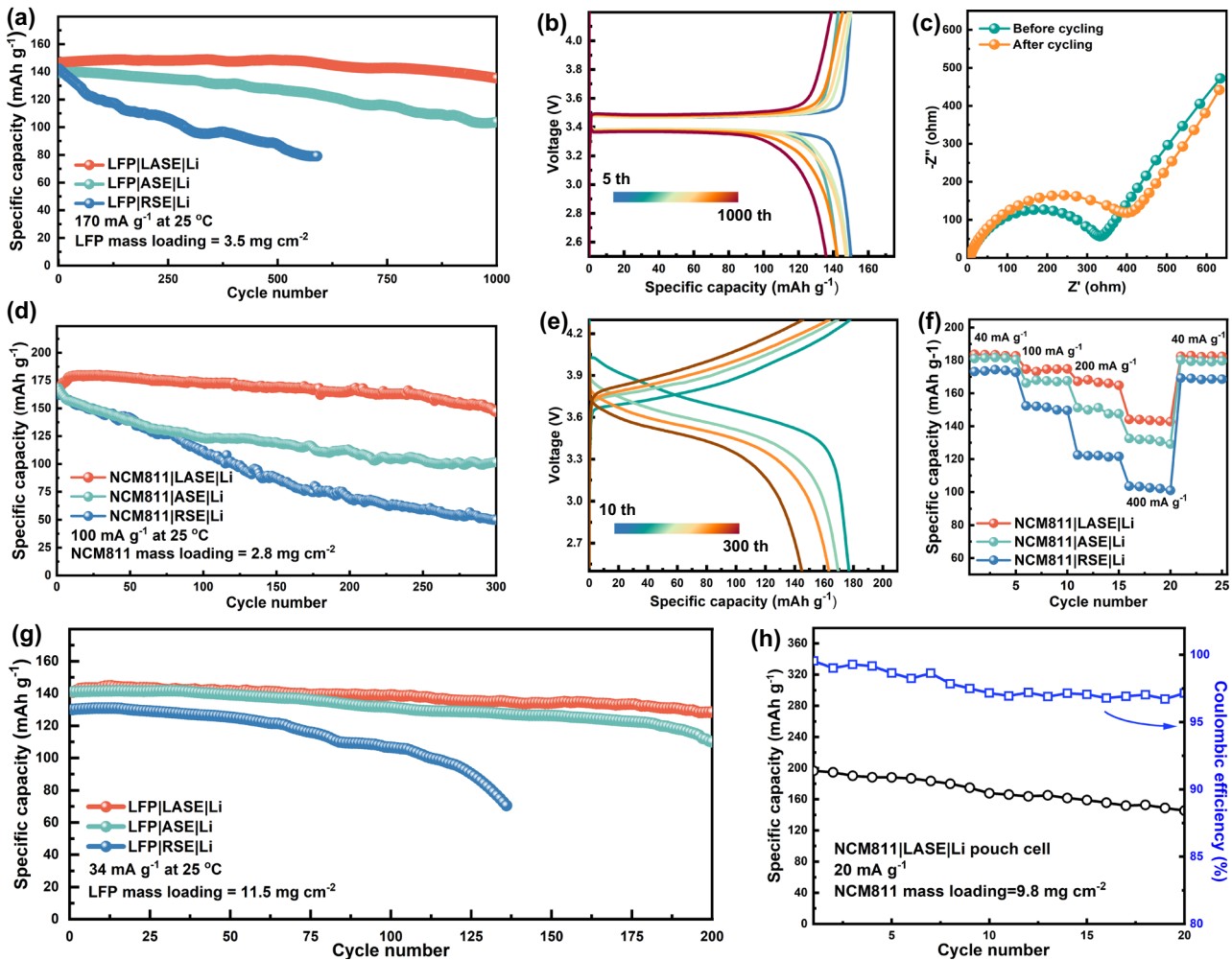

**Fig. 4 | Battery performances. a** Cycling performances of Li ||LFP cells using the LFP cathode of 3.5 mg cm⁻² mass loading. **b** Voltage profiles of the Li|LASE| LFP cell at various cycles. **c** EIS plots of Li|LASE| LFP before and after the cycling test. **d** Cycling performances of Li ||NCM811 cells. **e** Voltage profiles of the Li|LASE| NCM811 cell. **f** Rate capability of Li ||NCM811 cells. **g** Cycling performances of full cells assembled with the LFP cathode of 11.5 mg cm⁻² mass loading. **h** Cycling performances of the Li|LASE| NCM811 pouch cell.

about 600 cycles, whereas ASE delivered a prolonged cycling lifespan with a higher capacity retention of 74% after 1000 cycles. Among the studied electrolytes, the Li|LASE|LFP cell exhibited the highest initial capacity of 150.2 mAh g⁻¹ and capacity retention of 91%, accompanied by nearly 100% coulombic efficiency over 1000 cycles. Additionally, from voltage profiles (Fig. 4b and Supplementary Fig. 21), Li|LASE | LFP demonstrated a polarization voltage of 0.12 V at the 1000ᵗʰ cycle, much smaller than Li|RSE|LFP (0.36 V at the 500ᵗʰ cycle) and Li|ASE | LFP (0.21 V at the 1000ᵗʰ cycle). The Nyquist plots were tested to illustrate the mitigated polarization of Li|LASE | LFP. As shown in Fig. 4c and Supplementary Fig. 22, a drastic increment in the interfacial impedance was observed in Li|RSE|LFP and Li|ASE|LFP after cycling, suggesting a severely inhibited Li⁺ diffusion at the electrolyte-electrode interface, in contrast to the smaller resistance obtained in cycled Li|LASE|LFP which indicating a rapid kinetics. Li ||NCM811 cells with a NCM811 mass loading of 2.8 mg cm⁻² were assembled to estimate the high-voltage performance of the electrolytes. As presented in Fig. 4d and Supplementary Fig. 23, the initial capacities of Li ||NCM811 cells paired with RSE, ASE, and LASE were 159.8, 166.2, and 177.8 mAh g⁻¹ at the rate of at 100 mA g⁻¹. Li|LASE | NCM811 achieved higher capacity retention and much mitigated polarization voltage when compared to those of Li|RSE | NCM811 and Li|ASE | NCM811 (Figs. 4e and Supplementary Fig. 24). Apart from cycling ability, the rate capability of the

Li | |NCM811 cells was measured and compared in Fig. 4f. Compared to Li|RSE | NCM811 and Li|ASE | NCM811, the Li|LASE | NCM811 delivered much higher capacities of 182.0, 176.4, 168.2, and 144.1 mAh g⁻¹ at 40, 100, 200 and 400 mA g⁻¹, respectively. As illustrated in Supplementary Fig. 25 and Supplementary Table 5, compared to SPEs reported in recently published works[33,43–53], LASE still stands out, rendering good cycling performance in both Li | | NCM and Li ||LFP full cells.

Examining full cells paired with the high-mass loading cathode is imperative to validate the practicality. Firstly, Li||LFP cells were assembled with a high mass loading cathode (11.5 mg cm⁻²) and tested at 34 mA g⁻¹. As presented in Fig. 4g and Supplementary Fig. 26, Li|RSE|LFP had a capacity of 131.9 mAh g⁻¹ at the initial stage and exhibited a regretful capacity of 71.2 mAh g⁻¹ only after about 135 cycles. In contrast, Li|ASE|LFP and Li|LASE|LFP cells had higher initial capacities of 143.1 and 146.0 mAh g⁻¹, delivering ameliorated cycling performances with capacity retention of 76% and 88%, respectively (Supplementary Fig. 27). Ulteriorly, the Li| LASE|NCM811 punch cell with the cathode mass loading of 9.8 mg cm⁻² was assembled and measured at 20 mA g⁻¹. In Fig. 4h, after 20 cycles, the punch cell maintained a commendable capacity retention of 75% and can readily light up LED bubbles (Supplementary Fig. 28), whether in flat or folding states, manifesting the promising potential of LASE for practical application in

high-voltage LMBs. As revealed in Supplementary Table 6, though the LASE with 20 wt% of LLZTO showed a little higher areal density compared to ASE, the energy density of the LASE-based cell showed higher energy density due to the low mass proportion of LLZTO in the entire battery (~1.2 wt%) and higher discharge capacity/voltage of LASE-based cell.

## Mechanistic investigation of the interface stability

Considering the above results, the stable LASE-Li interface is one of the primary reasons for the improved battery performance. To deeply explore the enhancement mechanism of the interfacial stability concerned with the functional structures enabled by the supporting network, theoretical investigations were further implemented. The nanofibrous structures of PRN and APN were reconstructed via the X-ray nanotomography method. From the computed tomography (Fig. 5a), fibers in RPN were randomly arranged, in contrast to fibers in APN, which exhibited a noticeable improvement in orientation along the y-axis, demonstrating consistent results with SEM images in Fig. 2. The x-z surface perpendicular to the y-axis was regarded as the slicing plane to obtain cross-sectional images of RPN and APN. By comparison of the solid-line circles shown in Fig. 5a, cross-sections of fibers in RPN delivered irregular shapes, whereas cross-sections of fibers in APN appeared as relatively regular spots. Notably, the cross-sectional view of the fibrous mat partially reflects the pore distribution uniformity, which exerts a critical influence on regulating Li⁺ streaming. Accordingly, the mercury injection measurement was conducted to characterize the uniformity of porous features (Fig. 5b). It was found that pores in RPN and APN were mainly distributed at around 5 μm for the most intense peak. However, the dominant peak of RPN is broader and shorter than that of APN, and concurrently, a sub-peak was observed at around 25 μm, signifying the relatively uneven pore size distribution compared to APN.

Based on the structural analysis, the impact of fiber orientation on interfacial stability was investigated via modeling methods. Firstly, the phase-field equation fully coupled with Nernst-Planck-Poisson equations was employed to reproduce the transportation and deposition procedure of Li⁺, and the 2D corresponding computational domains were extracted from the reconstructed 3D models from the cross-sectional view. Unlike the previous phase-field methods for simulating the growth of Li dendrites[54–56], the model was developed without presetting nucleation sites at the initial phase boundary. Thus, the impact of the location and shape of dendrites was excluded from the present model. In this scenario, the supporting structure of electrolytes became the major factor influencing the dendrite evolution. Figure 5c and Supplementary Fig. 29, present the temporal-spatial propagation of the order parameter $\varphi$, the value of which distinguishes the electrode ($\varphi = 1$) and electrolyte phase ($\varphi = 0$). The outline of the interphase boundary was extracted using image processing technology to quantitatively characterize the morphology of the evolving dendrite by calculating the average discrepancy $R_a$ of the anode surface roughness. In the RSE model (Supplementary Fig. 30), the anode surface gradually evolved into a rugged surface as Li deposition proceeded, and an undesired $R_a$ value of 0.75 μm was accordingly obtained. Conversely, a relatively uniform deposition layer on the anode surface was observed in the ASE model (Supplementary Fig. 31), accompanied by a small $R_a$ value of 0.33 μm. The simulated results correspond to the dendric structures observed in SEM images in Fig. 2b and c. Apart from the superficial results by imitating the growth of Li dendrites, the distribution of Li⁺ concentration and over potential is further simulated by solving Nernst-Planck-Poisson equations. In Fig. 5d, the Li⁺ concentration ($c_{Li+}$) gradually increased along the direction away from the anode, thus leading to a concentration gradient upon the anode surface, which gave rise to Li dendrites preferentially growing from poor Li⁺ region to rich Li⁺ region for the balance of chemical potential[57]. Hence, the uniformity of the Li

deposition was primarily determined by the uniformity of $c_{Li+}$ distribution. In Supplementary Fig. 32, the tendency of the normalized $c_{Li+}$ along the horizontal section (white dash line) of RSE and ASE was obtained based on 2D simulation results. In this case, the fluctuation of $c_{Li+}$ distribution adjacent to APN was much smaller than that of RPN. The stack of aligned fiber arrays acted as shunting channels to guide the Li⁺ diffusion, leading to the uniform Li⁺ flux and thus contributing to a flat deposition layer. As visualized in Fig. 5e, the potential intensity of the anode surface was higher than that of the bulk electrolyte, rendering a similar distribution pattern with the $c_{Li+}$. In theory, the presence of the potential term in the Butler-Volmer equation (Eq. 8, Methods) suggests its significant impact on the kinetic of the electrochemical reaction. Thereby, the potential can be deemed as the driving force to promote the growth of dendrites, preferentially around the protuberant tips due to the charge enrichment phenomenon, as demonstrated in our previous investigations[58]. Significantly, the potential distributed in ASE exhibited upgraded uniformity and achieved an equipotential anode surface, allowing for the relatively smooth Li plating.

From the above simulations, the oriented structure of the PAN network can function as a regulator to homogenize the diffusion and reaction kinetics of Li⁺, thus leading to a stable interface in LMBs. Besides the contribution in the electrochemical aspect, numerical methods have also confirmed the mechanical suppression of Li dendrite growth by ASE. Figure 5f and g display the as-constructed 2D models, where the already formed Li dendrites were modeled as white metal balls, and electrolytes were modeled as composite bodies inserted in fibers with random and aligned orientations, representing RSE and ASE, respectively. Notably, the modeled RSE displayed more nonuniform pore size compared to ASE, which was consistent with the measured results of the pore size distribution in Fig. 5a and b. Accordingly, the penetration effect of Li dendrites on electrolytes was imitated by moving those metal balls toward the cuboid body. As the feeding displacement of balls was set as 2 μm, the maximum von Mises stress $\sigma_{max}$ in ASE was 261.7 MPa, much higher than that in RSE (71.2 MPa), exhibiting more efficient inhibition of Li dendrites at a mechanical level. At the same time, it was found that the $\sigma_{max}$ in ASE merely appeared at fibers, whereas the $\sigma_{max}$ in RSE was located at the conjunction of fibers, regarded as the weak point of the entire network. In sum, the oriented structure not only functions as a mechanical barrier to resist Li dendrites but also helps maintain the integrity of the supporting network, reducing the risk of being punctured by Li dendrites.

As demonstrated before, adding LLZTO particles reduced the activity between SN and Li anode. To clarify the mechanism, the calculation of the adsorption energy $E_{ads}$ was implemented based on density functional theory (DFT). The most stable surface of Li and LLZTO, assigned to (001) and (001) crystal planes[39], was constructed to adsorb SN, respectively. The optimized geometry of the adsorption configuration was exhibited in Fig. 5h and Supplementary Data 1. As shown in Supplementary Table 7, the $E_{ads}$ of Li-SN was calculated as -3.55 eV, indicating an intense activity between SN and Li metal. At the same time, the strong coordination of the LLZTO-SN pair was also verified, as the corresponding $E_{ads}$ reached up to -2.80 eV. Given their similar $E_{ads}$ value, LLZTO particles would compete with Li anodes in combination with SN molecules, thus reducing the activity of the Li-SN pair and finally leading to an electrochemically stable electrolyte-anode interface.

## Discussion

In summary, a specially engineered network composed of aligned PAN fibers loaded with LLZTO particles has been employed as the functional supporting skeleton to endow the composite SPE with high ionic conductivity, high uniaxial tensile strength, wide electrochemical window, high Li⁺ transference number and compatible interface with Li

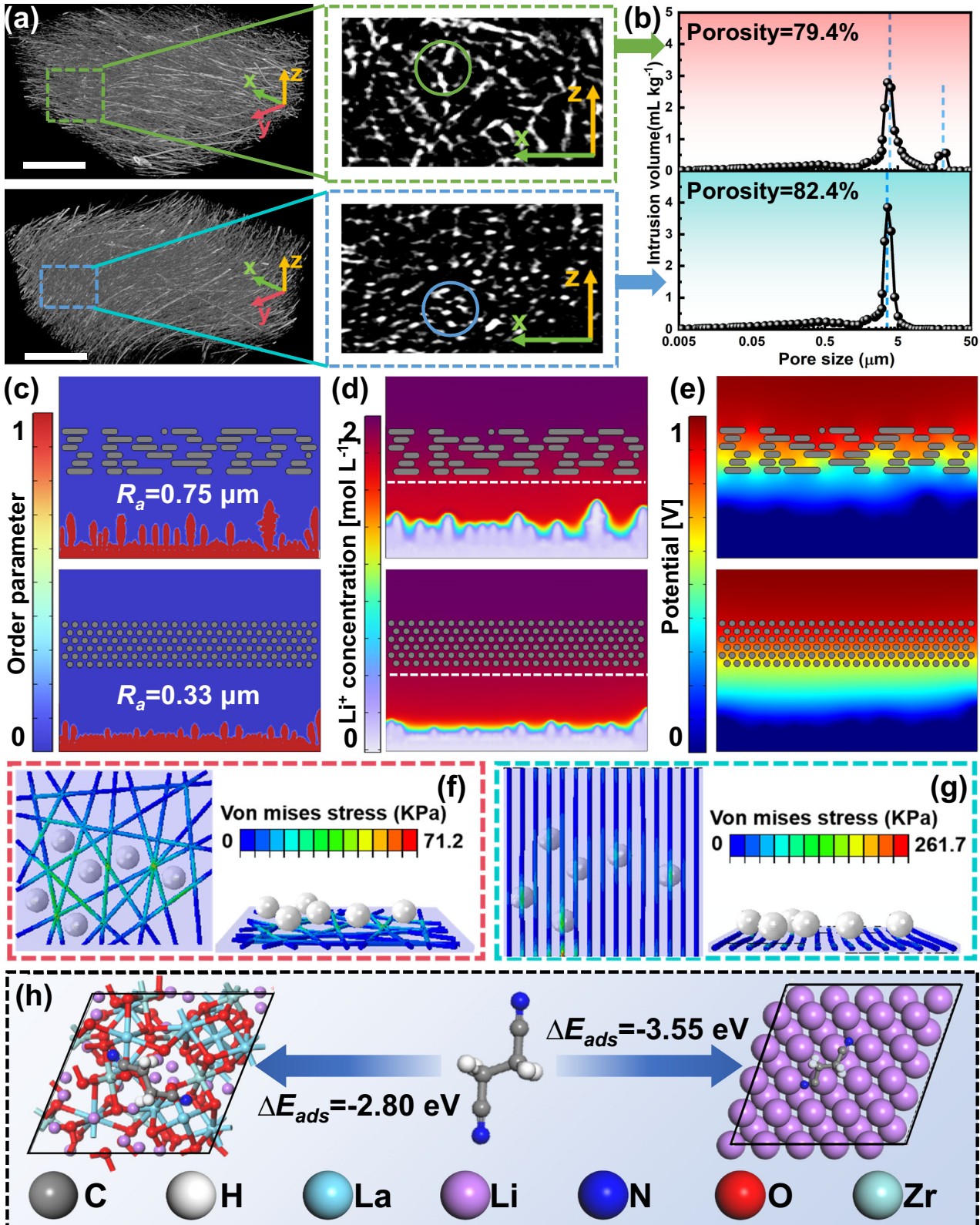

**Fig. 5 | Theoretical calculations. a** 3D-reconstructed images of RPN (up) and APN (down) by X-ray nanotomography. Scale bars, 50 μm. **b** Pore size distribution of RPN (up) and APN (down) by mercury injection. **c** Impact of the supporting structure on the growth of Li dendrites. **d** Simulated distribution of Li⁺ concentration. **e** Simulated distribution of electrical potential. **f, g** Stress distribution of electrolytes under the given displacement of Li dendrites (white balls: lithium dendrites, rods: PAN nanofibers): **f** RSE and (**g**) ASE. **h** Optimized configuration and calculated adsorption energy of LLZTO-SN and Li-SN.

anodes. Meanwhile, the Li plating/stripping procedure is observed as the steady polarization in the Li|LASE|Li cell applied with the current density of 0.5 mA cm$^{-2}$ over 1500 h. Consequently, such a stable electrolyte-anode interface gives rise to good cycling performances, of which the Li||LFP cell has a capacity retention of 91% after 1000 cycles, and the Li||NCM811 cell displays a capacity retention of 84% after 300 cycles. Significantly, as revealed by computational results derived from multiphysics simulations and DFT calculations, the tailored structure can function as 1) an ion-redistributor channeling Li$^{+}$ flux through the gaps among the aligned fibers, 2) a regulator uniforming the intensity of the electrical potential on the anode surface, 3) a physical barrier confronting Li dendrites at a mechanical level, and 4) a competitor with the Li anode in coordinating with SN, which collectively contribute to the stable interface in LMBs. The above results obtained from experimental and theoretical investigations confirm the effectiveness of the anisotropic concept in designing bi-functional SPEs, thus sharing valuable insights into developing flexible energy devices.

## Methods

### Materials

All reagents are commercially available and used as received without further purification. Li$_2$CO$_3$ (99.9%), La$_2$O$_3$ (99.9%), ZrO$_2$ (99.9%), Ta$_2$O$_3$ (99.9%), poly(ethylene glycol) diacrylate (PEGDA, M$_w$ = 400 g mol$^{-1}$), succinonitrile (SN, 99%), and lithium bis(trifluoromethanesulphonyl)imide (LiTFSI, 99.9%), 2,2′-Azobis(2-methylpropionitrile) (AIBN, 99.9%), DMF (N,N-Dimethylformamide, 99.9%), NMP (N-Methylpyrrolidone, 99.9%) were purchased from Aladdin Ltd. (Shanghai, China). Polyacrylonitrile (PAN, M$_w$ = 150000 g mol$^{-1}$) was purchased from Sigma-Aldrich. LiFePO$_4$ (LFP), LiNi$_{0.8}$Co$_{0.1}$Mn$_{0.1}$O$_2$ (NCM811), Li foil (99.9%, 40 μm thickness), polyvinylidene difluoride (PVDF), the high-loaded positive electrode sheets (LFP electrode:11.5 mg cm$^{-2}$, 65 μm thickness; NCM811 electrode: 9.8 mg cm$^{-2}$, 40 μm thickness), aluminum tabs, nickel tabs and aluminum-plastic film were were purchased from Guangdong Canrd New Energy Technology Co., Ltd.

### Preparation of Li$_{6.4}$La$_3$Zr$_{1.4}$Ta$_{0.6}$O$_{12}$ particles

The nano-sized Li$_{6.4}$La$_3$Zr$_{1.4}$Ta$_{0.6}$O$_{12}$ (LLZTO) particles were synthesized by the conventional solid-state reaction and high-energy ballmilling. The starting materials consisted of stoichiometric amounts of Li$_2$CO$_3$, ZrO$_2$, Ta$_2$O$_5$, and La$_2$O$_3$, with 10 wt.% excess Li$_2$CO$_3$ added to compensate for lithium loss during the sintering process. The raw materials were milled in isopropanol at a speed of 500 rpm for 12 h, maintaining a mass ratio of isopropanol to raw materials of 3:1. After ball milling, the precipitate was collected by centrifugation and dried in a vacuum oven at 60 °C for further use. The dried mixture was sintered at 900 °C for 12 h and the powder was collected after cooling.

### Preparation of fiber networks

The functional fiber networks were prepared through the electrospinning process. In detail, PAN and LLZTO were separately dispersed uniformly in DMF solution. After stirring each solution for 2 h, the two solutions were mixed. Following an additional 0.5 h of stirring, electrospinning was promptly conducted to prevent LLZTO aggregation. A high DC voltage of 15 kV and the tip-to-collector distance of 15 cm were adopted during the electrospinning process. After setting the rotational speed of the drum collector to 800 rpm and 4500 rpm, random and aligned fibers were respectively collected on the attached aluminum foil. 20 wt% LLZTO nanofillers (based on the mass of PAN and LLZTO) were added to the precursor solution and vigorously stirred for 4 h to obtain a homogeneous spinning solution. The aligned LLZTO/PAN fiber arrays were prepared via a similar electrospinning method, as the drum rotating speed was configured to 4500 rpm. After drying in a vacuum oven at 70 °C overnight, the obtained fibrous

separators were peeled off from the collecting paper and tailored into a 16 mm diameter film for the further use.

### Preparation of solid polymer electrolyte

Solid polymer electrolytes are prepared by in situ curing. The precursor solution for the in situ curing was prepared by mixing PEGDA, SN, and LiTFSI into a transparent solution at the weight ratio of 0.3:1.2:0.95. Meanwhile, 1 wt% AIBN was added into the precursor to initiate the polymerization of PEGDA segments. For the assembly of Li symmetrical cells, the supporting skeleton was sandwiched between two SS discs, and then 30 μL of the precursor solution was drawn using a pipette and injected into the porous network. After sealing the coin cell, the Li||Li symmetrical cells were transferred into an oven and heated at 70 °C. Likewise, the stainless steel (SS) symmetrical cell was fabricated by replacing the two Li anodes with SS discs, and the Li||SS asymmetrical cell was assembled by replacing one of the Li anodes with SS discs. The Li||LFP and Li||NCM811 full cells were assembled by replacing the two Li anodes with LFP and NCM811 films in a similar in-situ method, respectively.

### Characterizations

Scanning electron microscopy (SEM, Zeiss Gemini 300) was used to observe the surface and cross-sectional morphology of supporting networks and solid electrolytes. The pores size distribution and porosity of the supporting network were obtained by mercury injection technology (AutoPore 9500). The crystalline structures of the studied networks were identified by Wide-angle X-ray scattering and small-angle X-ray scattering (WAXS and SAXS, Xeuss 2.0) analysis. Ex situ X-ray photon-electron spectroscopy (XPS, Thermo Kalpha) was performed to detect the chemical information of the Li negative electrode surface. The microstructures of RPN and APN were reconstructed by 3D nanotomography technology (Versa 620, Zeiss) with an accelerating voltage of 80 KV. To prepare samples for mechanical measurements, the supporting networks were first put on a glass substrate (thickness of 0.5 mm), then the precursor solution was injected into the skeletons. Subsequently, another glass plate was placed on the top of the wetted network. After sufficiently polymerizing at 70 °C, the membrane samples were peeled off from glass substrates. To estimate the tensile strength of the entire electrolyte, the samples were fixed in a clamp and stretched by a universal testing machine (WDW-2) at a feeding speed of 2 mm min$^{-1}$. Correspondingly, Young's modulus can be calculated from the initial slope-linear stage of the stress-strain curves.

### Electrochemical measurements

The mentioned electrochemical performances of electrolytes were measured by the electrochemical workstation (Gamry, Interface 5000E). The ionic conductivity (σ) was evaluated by EIS plots of SS||SS (15.6 mm) cells as the frequency varies from 100 kHz to 0.1 Hz. The σ value was calculated according to the equation:

$$\sigma = \frac{L}{R_b \times A} \tag{1}$$

where $A$ (cm$^2$) is the effective contact area (1.91 cm$^3$) between the electrolyte and electrode. $L$ (cm) is the thickness of the electrolyte. $R_b$ is the bulk resistance derived from the EIS. The formula to calculate the activation energy ($E_a$) is derived from the Arrhenius equation:

$$\sigma = A \exp(E_a/RT) \tag{2}$$

where $A$ is the pre-exponential factor, $T$ is the absolute temperature and $R$ is the gas constant (8.3145 J mol$^{-1}$ K$^{-1}$).

The Li$^+$ transference number ($t_+$) of the electrolyte was measured according to the Bruce-Vincent-Evans equation:

$$t_{Li^+} = \frac{I_S(\Delta V - I_0 R_0)}{I_0(\Delta V - I_S R_S)} \quad (3)$$

where $\Delta V$ is maintained at 10 mV during testing. $I_0$ and $R_0$ correspond to the initial current and initial interfacial resistance, while $I_s$ and $R_s$ denote the stabilized current and resistance values after prolonged polarization. These parameters can be directly extracted from electrochemical characterization data obtained through chronoamperometric testing and electrochemical impedance spectroscopy (EIS) performed on Li-Li symmetric cells[59].

For the fabrication of electrodes, the positive electrode slurry was prepared by mixing the positive electrode active material (LFP,NCM), conductive carbon black, PVDF in a weight ratio of 8:1:1 using NMP as the solvent in a mortar with pestle. The resulting slurry was coated onto aluminum foil with a doctor blade in air and dried in a vacuum at 80 °C for 12 h. The loading amount of the positive electrode was controlled by adjusting the thickness of the doctor blade. Then, the positive electrode sheets were punched into circular electrodes with a diameter of 12 mm and dried again overnight at 120 °C in a vacuum oven to remove residual solvents and moisture. For coin cells, all were assembled using the 2032-type coin cells, which did not require additional pressure during electrochemical measurements. For pouch cells, the dimension of positive electrodes and Li negative electrode was rectangular (30 mm × 40 mm). aluminum tabs were employed on the positive electrode side, while nickel tabs were used on the negative electrode. To prevent short circuits caused by misalignment during manual stacking, the separator/solid polymer electrolyte was designed to be slightly larger than the electrodes, with dimension of 32 mm × 42 mm. Similar to the coin cell assembly, 180 μL of the precursor solution was drawn using a pipette for in situ polymerization. During this process, the pouch cell was sandwiched between two alumina plates to ensure adequate interfacial contact. After polymerization was completed, the pouch cell was subjected to secondary packaging to expel any residual gases from the interior. Additionally, pouch cells were subjected to an external pressure of ~100 kPa during cycling tests. In coin cell tests, the positive electrode loading amounts for LFP and NCM are typically 3.5 mg cm$^{-2}$ and 2.8 mg cm$^{-2}$, respectively. For stability testing, an LFP positive electrode with a high loading of 11.5 mg cm$^{-2}$ was employed. Additionally, an NCM positive electrode with a loading of 9.8 mg cm$^{-2}$ was used for the assembly of pouch cells. All electrochemical measurements were conducted at a constant temperature of 25 °C in a thermostatic chamber. All cell assembly was carried out in a glove box ($H_2O$ < 0.1 ppm and $O_2$ < 0.1 ppm).

The electrochemical stability window of the solid electrolyte was detected by the linear sweep voltammetry (LSV) method performed from the open circuit voltage to 5.5 V or -0.2 V a scanning rate of 1 mV s$^{-1}$. The Li depositing/stripping behavior was estimated by voltage profiles, which are derived from the D.C. galvanostatic cycling of symmetrical Li||Li cells at current densities of 0.5 and 1 mA cm$^{-1}$. The electrochemical impedance spectra (EIS) of the cells were measured by the constant potential method with a frequency range of 0.1 HZ to 10$^6$ HZ and an AC amplitude of 10 mV. Prior to carry out the EIS test, all cells were left in open circuit voltage for 30 mins. EIS data were fitted using an equivalent electrical circuit with ZView software. The battery performances were evaluated by rate capability and cycling performance of full cells using battery test instruments (LANHE, CT3002A). For full cells paired with the low-mass-loading LFP positive electrode, each cycle was performed in the potential window ranging from 2.5 V to 4.2 V at 170 mA g$^{-1}$. For full cells equipped with the low-mass-loading NCM811 positive electrode, each cycle was performed in the potential window ranging from 2.5 V to 4.3 V at a 100 mA g$^{-1}$. As for full cells with the high-mass-loading LFP positive electrode, the current rate was set

at 34 mA g$^{-1}$. The cycling performance of the Li||NCM811 pouch cell was measured at 20 mA g$^{-1}$. Coulombic efficiency is defined as the ratio of discharge capacity divided by the charge capacity in the preceding charge cycle.

## Simulation of Li$^+$ migrating behavior

The transport trace of Li$^+$ in electrolytes was imitated by Transport of Diluted Species modules in COMSOL Multiphysics 6.1. The physical model is governed by the second Fick's law:

$$\frac{\partial c}{\partial t} = \frac{\partial}{x}\left(D_x \frac{\partial c}{\partial x}\right) + \frac{\partial}{y}\left(D_y \frac{\partial c}{\partial y}\right) \quad (4)$$

In this model, $D_x$ is equal to $D_y$, which can be derived from the ionic conductivity based on the Nernst-Einstein equation. The 2D computational domains were distinguished by the morphology of the supporting networks. In the case of RSE, its cross-section can be defined as a rectangle embedded with randomly distributed ellipses due to the disordered fibrous structure of RPN. For ASE, the cross-section of the inserted network is converted into uniformly distributed circles because of the highly oriented fiber arrays. As for LASE, to describe the combination of aligned fibers and inorganic particles, the model was characterized as uniformly distributed circles with alternating sizes; the larger corresponded to the cross-section of LLZTO particles, and the smaller corresponded to the cross-section of PAN fibers. The overall domain was configured as a rectangle of 250 (width) × 80 (thickness) μm$^2$.

## Influence of Li$^+$ transference number

The distribution of Li$^+$ stimulated by electric fields was imitated based on Electrostatic and Transport of Diluted Species modules combined in COMSOL Multiphysics. The modeling method is based on Nernst-Planck-Poisson equations:

$$\frac{\partial c_i}{\partial t} = -\nabla \boldsymbol{N}_i \quad (5)$$

$$\boldsymbol{N}_i = -D_i \nabla c_i + \mu_i c_i \boldsymbol{E} \quad (6)$$

$$\boldsymbol{E} = -\varepsilon_o \varepsilon_r \nabla \quad (7)$$

where $V$ is the overpotential, $E$ is the electric field, $D_i$ is the diffusion coefficient of species i, $c_i$ is the concentration of species i, $\mu_i$ is the ionic mobility of species i, and $Ni$ is the flux vector of species i. As the driving force for Li$^+$ transport, the overpotential difference between the anode and cathode was set as 0.01 V. The Li$^+$ flux on the boundary, which resulted from the electrochemical reaction, was set as 0.025 mol m$^{-2}$ s$^{-1}$. The diffusion coefficient of each species is derived from the ionic conductivity based on the Nernst-Einstein equation. For the comparison of electrolytes with various Li$^+$ transference numbers, the computational domain was configured as a rectangle with the size of 45 (thickness) × 15 (width) μm$^2$. According to the examined Li$^+$ transference number, anions were divided into two sets: one corresponds to the free anions with the normal diffusion coefficient, and the other corresponds to the immobilized anions with a diffusion coefficient of 0.

## Modeling of the Li dendrite propagation

The computational domain is divided into Li anode, PEGDA/SN electrolyte, and PAN fiber, respectively. When the battery is charged, Li$^+$ moves towards the anode driven by the overpotential. As a result, Li$^+$ reacts with electrons and is reduced to Li on the electrode surface. If the electrodeposition is inhomogeneous, the growth of lithium dendrites will be induced. The Phase-field model (PFM) has been

employed to simulate the complex morphologies resulting from dendritic growth, leveraging its ability to accurately and efficiently describe the evolution of interfaces[60]. The corresponding computational domains were extracted from the reconstructed 3D models from the cross-sectional view. Considering the author's former research[58], the time-dependent interface evolution is expressed as follows:

$$\frac{\partial \varphi}{\partial t} = -L_i \left[ \frac{\partial g(\varphi)}{\partial \varphi} - \kappa \nabla^2 \varphi \right] - L_m h'(\varphi) \left\{ e^{\left[ \frac{(1-\alpha)nFP}{RT} \right]} - \frac{c_+}{c_0} e^{\left[ \frac{-\alpha nFP}{RT} \right]} \right\} \quad (8)$$

where $\varphi$ is the order parameter, of which the value continuously varies from 1 to 0 in the interphase. $L_i$ and $L_m$ are the interfacial mobility and reaction coefficient, respectively. $g(\varphi) = W\varphi^2(1-\varphi)^2$ is the double-well function. $\kappa$ is the gradient energy coefficient expressed by $\kappa_0(1 + \delta \cos \omega\theta)$, where $\delta$ and $\omega$ are the strength and mode of the EEI anisotropy. $h(\varphi) = \varphi^3(6\varphi^2 - 15\varphi + 10)$ is an interpolation function. $R$, $T$, $n$ and $F$ are the gas constant, temperature, number of the transfer electrons, and Faraday's constant, respectively.

From the above equations, $\varphi$ is coupled with the Li$^+$ concentration $c^+$ and overpotential $P$. Nernst-Planck equation is written as follows to simulate the temporal and spatial distribution of Li$^+$ concentration:

$$\frac{\partial c^+}{\partial t} = \nabla \cdot \left( D_{eff} \nabla c^+ - \frac{D_{eff} nF}{RT} \nabla P \right) - c_{Li} \frac{\partial \varphi}{\partial t} \quad (9)$$

where $D_{eff}$ is the effective diffusivity, interpolated as $D_e \, h(\varphi) + D_l \, [1-h(\varphi)]$. Here, $D_e$ and $D_l$ are the diffusivity of Li$^+$ in the electrode and liquid electrolyte, respectively. Further, Poisson's equation is used to ensure the electric neutrality of the system:

$$\nabla \cdot \sigma_{eff} \nabla P = nF c_{Li} \frac{\partial \varphi}{\partial t} \quad (10)$$

where $\sigma_{eff}$ is the effective conductivity, written as $\sigma_e h(\varphi) + \sigma_l [1-h(\varphi)]$, where $\sigma_e$ and $\sigma_l$ correspond to the conductivity of Li$^+$ in the electrode and liquid electrolyte, respectively. The mentioned governing equations are solved by the finite element method on COMSOL Multiphysics 6.1. The critical parameters are listed in Supplementary Table 8.

## DFT calculation
All computations were performed within the density functional theory (DFT) framework, utilizing the Perdew-Burke-Ernzerhof (PBE) exchange-correlation functional under the generalized gradient approximation (GGA). The CASTEP code implemented in Materials Studio was employed for these simulations. A plane-wave basis set with an energy cutoff of 550 eV was selected to expand the electronic wavefunctions across all k-point sampling. For Brillouin zone integration, a Gamma-centered Monkhorst-Pack k-point grid with $3 \times 3 \times 1$ mesh configuration was implemented to ensure accurate convergence[61]. Geometries were optimized until the energy $(1.0 \times 10^{-6} \text{ eV/atom})$ and force $(0.01 \text{ eV/Å})$ converged[62]. Geometries were optimized until the energy was converged to $1.0 \times 10^{-6}$ eV/atom, and the force was converged to 0.01 eV/Å. The LLZTO bulk structure with space group Ia$3d$ was originally taken from the International Crystal Structure Database (ICSD)[63].

The adsorption energy (Eads) of SN-LLZTO (001) and SN-Li (001) surface are calculated as follows: $E_{ads} = (E_{adsorbate+sub} - E_{sub} - E_{adsorbate})$.

## Data availability
The data that support the findings detailed in this study are available in the article and its Supplementary Information or from the corresponding authors on request. The X-ray crystallographic coordinates for structures reported in this study have been deposited at the Cambridge Crystallographic Data Centre (CCDC). Source data are provided with this paper.

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

## Acknowledgements

The authors gratefully acknowledge the financial support from the Key-Area Research and Development Program of Guangdong Province (2023B0909050008 (S.W.)), and the National Natural Science Foundation of China (22408205 (J.G.), 22408198 (Z.Z.), 22178120 (S.W.) and 22478127 (S.W.)). This work is partially supported by High Performance Computing Platform of South China University of Technology.

## Author contributions

J.G. contributed to the design of the research and performed the experimental data analysis. J.G. and Z.Z. conducted the materials synthesis, electrochemistry, and cell performance. J. Gou and K. Cui performed the material characterizations and electrochemical measurements. J. Gou and J. Huang conducted the theoretical calculation and related data analysis. Z.Z., S.W. and H.W. supervised the work. All authors discussed the results, co-wrote, and commented on the manuscript.

## Competing interests

The authors declare no competing interests.
