## [Transparent Peer Review file · Nature Communications]

An Anisotropic Strategy for Developing Polymer Electrolytes Endowing Lithium Metal Batteries with Electrochemo-mechanically Stable Interface

Corresponding Author: Professor Haihui Wang

Version 0:

Reviewer comments:

Reviewer #1

(Remarks to the Author)

In this manuscript, the author claims to have achieved a highly anisotropic new concept polymer electrolyte by combining PAN fibers modified with LLZTO particles with the polymer matrix. Beneficial from the highly anisotropic nature of the composite solid-state electrolyte and the synergistic effects of multiple components, effective control over the chemical reactions between the lithium metal anode and the composite solid-state electrolyte is achieved, along with the suppression of lithium dendrite formation and resistance to lithium dendrite penetration through the electrolyte. The author has designed a solid polymer electrolyte with high ionic conductivity and outstanding ability to resist lithium dendrites through clever strategy. However, further elucidation of certain underlying mechanisms is still need to be strengthened. A major revision is recommended.

Q1: The schematic diagram (Figure 1a) doesn't explain clearly how the polymer matrix is combined with LASE, and there's no indication of how increasing the rotation speed of the collector was employed to achieve a network (APN) with the upgraded orientations. An illustrative diagram with essential step-by-step explanations is crucial for enabling readers to quickly grasp the design concept.

Q2: The author selected PAN as the host material for the solid-state polymer electrolyte in this study. However, there are numerous other polymers available for selection. So, what are the unique advantages of the utilization of PAN?

Q3: The part of the manuscript that uses SAXS to analyze the distributing orientation of PAN fibers fails to explain why the presence of an elliptical pattern indicates that PAN is uniaxially aligned. The author should provide relevant evidence in the part.

Q4: The nanoscale LLZTO particles tend to aggregate within the polymer, resulting in uneven distribution within the polymer matrix. As a consequence, this uneven distribution may lead to uneven distribution of particles on PAN fibers, potentially influencing the migration of lithium ions. However, this relevant aspect is not addressed in the manuscript.

Q5: The good wettability of the polymer on the lithium metal anode is also crucial, but there is no relevant performance characterization provided in the manuscript.

Q6: PEGDA is a polymer with two C=C end bonds and an ethylene oxide segment (-CH₂-CH₂-O-, EO) in the middle. Though SPEs containing EO segments usually exhibit good ionic conductivity, EO segments start to be oxidized at a voltage of over 3.9 V (vs. Li/Li⁺). However, the electrochemical performance of NCM811||Li cells showed no deterioration. Experimental data on the resistance of the SPE to NCM811 oxidation should be provided.

Q7: Compared with other reported strategies for SPE and inorganic SSE, (for example, Energy Storage Materials, 2023, 63, 103009, Energy Storage Materials, 2022, 50, 810, Journal of Colloid and Interface Science, 2023, 642, 193, and Nano Energy, 2022, 91, 106643), what's the advantage and disadvantage of the LAPN-based SE from the aspects of interface property and electrochemical performance?

Reviewer #2

(Remarks to the Author)

This paper introduced a novel anisotropic design for a solid polymer electrolyte (SPE), aiming to enhance its ionic conductivity and stability. The study incorporates various experimental and theoretical analyses, including SEM, X-ray scattering measurements, EIS, electrochemical and mechanical measurements, XPS, as well as modeling and calculations. Through these analyses, the authors demonstrate that the prepared polyacrylonitrile (PAN) matrix with an oriented structure shows superior properties compared to the conventional randomly distributed one. While this work has the potential to contribute to the development of solid-state lithium metal batteries (LMBs) with SPEs, further enhancement in novelty and scientific rigor as well as significant revisions to the structure and flow of the paper are required to meet the standards for publication in Nature Communications. Therefore, based on the feedback provided below, I advise against publishing this manuscript.

1. In the Introduction, it would be beneficial to provide the full names for PEGDA, SN, and LiTFSI to aid readers who may not be familiar with these abbreviations. Review other terminologies to ensure they are also clearly specified.
2. Can the authors provide the rationale for selecting these materials for the LAPN electrolyte, namely PAN, PEGDA, SN, LiTFSI, and LLZTO, instead of other alternatives commonly used in solid polymer electrolytes (SPEs)? This explanation would ideally be placed at the beginning of the Results for clarity of the context.
3. Distinguishing the difference between the WAXS data of RPN and APN in Figure 1d is challenging. To enhance the visibility of any differences between the two samples, the authors are recommended to plot the intensities of the data.
4. The SEM image for LAPN in Figure 1f does not provide conclusive evidence regarding the successful combination of PAN and LLZTO. To address this limitation, the authors are recommended to conduct elemental analysis, such as energy-dispersive X-ray spectroscopy (EDS), to provide further insight into the composition of the material.
5. The 2D diffusion models for RSE, ASE, and LASE in Figure 2b and Supplementary figure 7 should be rigorously demonstrated to accurately reflect their real microstructures and to avoid biased comparisons. Specifically, the total cross-sectional area of PAN fiber should remain constant for each model.
6. The XPS results in Fig. 3d and e and their explanations require several revisions: 1) Two peaks at 685.1 (LiF) and 688.4 eV (TFSI⁻) in the F 1s spectra is conversely corresponded as TFSI⁻ and LiF, respectively. 2) Reverse the direction of the x-axis, with lower binding energy to the right. 3) The peak deconvolution appears to have been done somewhat inaccurately. The FWHM and energy values among identical components should match more closely than they currently do. I would recommend redoing the peak deconvolution, as proper interpretation will only be possible thereafter.
7. The cycle performances of SPEs are evaluated as shown in Figure 4 by pairing them with LFP and NCM cathodes, and Li metal anodes. The results indicate that LASE outperformed ASE; however, with the 20 wt% of LLZTO in LASE compared to ASE, there might be a loss of energy density in the cell as a trade-off for better cyclability. I recommend that the authors to calculate and compare the energy densities (in Wh/kg and Wh/L) of the cells with LASE and ASE.

Version 1:

Reviewer comments:

Reviewer #1

(Remarks to the Author)

In my opinion, the manuscript is well-organized and the authors have given sufficient answers to the Reviewers' comments. Therefore, the manuscript can be published in this journal.

Reviewer #2

(Remarks to the Author)

The authors have thoroughly addressed all comments, and the changes made have improved the quality of the manuscript. Their responses to some of my comments have been satisfactory, however, not all issues from the first review have been resolved.

The authors added more analysis as I suggested, but the evidence for the results remains arguable, making the discussions controversial. The EDS and TEM images in Supplementary Figure 2 and 3 do not provide further evidence. It is recommended to conduct a more robust experiment to identify the successful combination of PAN and LLZTO. The XPS data in Figures 3d and e were displayed in the correct format, but the deconvoluted peaks and their explanations are still unclear; further experiments are required to find the role of LLZTO.

Overall, the authors provide a comprehensive explanation of the aligned PAN, but they could not adequately elucidate on the reasonings regarding LLZTO. A more robust analysis or stronger evidence is required to improve the reliability of this work. Currently, the manuscript seems insufficient for publication.

Version 2:

Reviewer comments:

Reviewer #2

(Remarks to the Author)

The authors have thoroughly addressed all the concerns raised by the reviewer in the revised manuscript, and it is now recommended for publication.

Title: An Anisotropic Strategy for Developing Polymer Electrolytes Endowing Lithium Metal Batteries with Electrochemo-mechanically Stable Interface

Authors: Jingren Gou, Zheng Zhang, Kaixun Cui, Suqing Wang, Jiale Huang, Haihui Wang

Many thanks to the reviewers for their valuable comments and suggestions. The following are the point-by-point responses to the concerns:

Response to Reviewer #1

Original comments:

Comment 1: The schematic diagram (Figure 1a) doesn't explain clearly how the polymer matrix is combined with LASE, and there's no indication of how increasing the rotation speed of the collector was employed to achieve a network (APN) with the upgraded orientations. An illustrative diagram with essential step-by-step explanations is crucial for enabling readers to quickly grasp the design concept.

Response: Thank you very much for your comment. According to your suggestion, we have revised the schematic diagram and shown in Figure 1a.

Figure 1. Structure characterizations. a Schematic diagram of the preparation procedure. b, c SEM images of surface morphology: b RPN and c APN. Scale bars, $1\mu\text{m}$. d, e WAXS and SAXS patterns. f Surface morphology of electrolytes of LAPN. Scale bar, $1\mu\text{m}$. g Mechanical properties of the proposed electrolytes.

Comment 2: *The author selected PAN as the host material for the solid-state polymer electrolyte in this study. However, there are numerous other polymers available for selection. So, what are the unique advantages of the utilization of PAN?*

Response: Thank you very much for your comment. PAN was chosen for the following reasons. The principal chain architecture of PAN encompasses a C≡N polar group, serving as a Lewis base capable of coordinating with lithium ions and dissociating lithium salts. This C≡N polar group possesses a pronounced electron-withdrawing capacity, rendering PAN with a low LUMO and exceptional electrochemical stability (J. Am. Chem. Soc., 2019, 141, 23, 9165–9169; Energy Storage Mater., 2022, 51, 130-138; Energy Storage Mater., 2021, 34, 388-416; J. Membr. Sci., 2015, 486, 21-28; J. Power Sources, 2011, 196, 6742-6749). Consequently, PAN is demonstrated as an efficient host matrix that can impart good anti-oxidation characteristics to the electrolyte. Furthermore, the interface between PAN and LLZTO may foster intimate chemical interactions, promoting partial dehydrogenation of PAN and yielding localized conjugated structures. These conjugated structures enhance Li⁺ interactions and expedite Li⁺ exchange at the LLZTO/PAN interface, thereby establishing a contiguous Li⁺ conduction pathway and fostering uniform Li deposition (Solid State Ionics, 2018, 327, 32-38; J. Am. Chem. Soc., 2021, 143, 5717-5726).

According to your suggestion, the merits of PAN were added and the relevant references have been cited in the revised manuscript (Page 4, line 93-95).

“As revealed in previous studies, both of PAN and SN encompass C≡N polar group, which are demonstrated to contribute to enhancing the electrochemical window of electrolyte.^{25,26}”

Comment 3: *The part of the manuscript that uses SAXS to analyze the distributing orientation of PAN fibers fails to explain why the presence of an elliptical pattern indicates that PAN is uniaxially aligned. The author should provide relevant evidence in the part.*

Response: Thank you very much for your comment. SAXS technique is one of the

most significant methods for characterizing the crystalline orientation of polymeric materials due to its statistical average and nondestructive detecting feature. A monochromatic parallel beam of X-rays is used for the scattering experiment. As X-rays pass through a sample, the oscillating electromagnetic field of X-rays interacts with electrons, making the electrons secondary sources of X-rays of the same frequency. Those secondary X-rays interfere with each other to form a specific pattern deviating from the primary beam path depending on the actual locations of the electrons in the sample. Therefore, if the polymer network has a random structure, the scattering pattern correspondingly exhibits a relatively uniform circle due to the random beam path. In contrast, if the polymer network has an aligned fibrous structure, the beam path is relatively oriented, thus leading to an oriented scattering pattern, such as the elliptical pattern obtained in this manuscript (Polymer, 2012, 53, 4702-4708; European Polymer Journal, 2013, 49, 33-40; Nanoscale Adv., 2022, 4, 491-501; Macromolecules 2005, 38, 3883–3893).

Figure 1e SAXS patterns of RPN and APN.

According to your comments, the corresponding description and relevant literature citation have been added in the revised manuscript (Page 5, line123-125).

“Wide-angle X-ray scattering (WAXS) and small-angle X-ray scattering (SAXS) measurements were conducted to analyze the distributing orientation of PAN fibers at a statistical level.^{28, 29}”

Comment 4: *The nanoscale LLZTO particles tend to aggregate within the polymer, resulting in uneven distribution within the polymer matrix. As a consequence, this uneven distribution may lead to uneven distribution of particles on PAN fibers, potentially influencing the migration of lithium ions. However, this relevant aspect is not addressed in the manuscript.*

Response: Thank you very much for your comment. In general, the addition of high-content LLZTO tends to cause gelation of the PAN/DMF solution, thus leading to the uneven distribution of LLZTO particles on PAN fibers. In this study, the homogeneous mixing of PAN with LLZTO was primarily achieved by optimizing the precursor solution preparation for electrospinning. Initially, PAN and LLZTO were separately dispersed uniformly in the DMF solution. After stirring each solution for 2 hours, the two solutions were mixed. Following an additional 0.5 hours of stirring, electrospinning was promptly conducted to prevent LLZTO aggregation. This process ensured the homogeneous distribution of LLZTO nanoparticles in the electrospun fibers of PAN. According to your comments, the corresponding description has been added in the Method section of the revised manuscript (Page 16, line 443-447).

“In detail, PAN and LLZTO were separately dispersed uniformly in the DMF solution. After stirring each solution for 2 hours, the two solutions were mixed. Following an additional 0.5 hours of stirring, electrospinning was promptly conducted to prevent LLZTO aggregation.”

Comment 5: *The good wettability of the polymer on the lithium metal anode is also crucial, but there is no relevant performance characterization provided in the manuscript.*

Response: Thank you very much for your comment. According to your suggestion, we have conducted the contact angle measurement to evaluate the wettability of the polymer precursor on the lithium metal anode (as shown in Figure S5). The excellent wettability of the polymer precursor on the lithium metal anode contributes to the close contact between SPEs and Li anodes after in-situ polymerization, which is beneficial to

reducing the internal resistance of LMBs. The close contact between the as-designed SPE and Li anodes was validated by cross-sectional SEM images of Li/Li symmetrical cells (Figure S8).

According to your suggestion, we have added the related description in the revised manuscript (Page 6, line 146-148, line 158-159).

“Prior to the curing process, the affinity of polymer precursor on the lithium anode was validated via the contact angle measurement (Supplementary Fig. 5).”

“Significantly, the close contact between the as-designed SPE and Li metals was validated by SEM images of Li/Li symmetrical cells (Supplementary Fig. 8).”

Supplementary Figure 5. Contact angle measurement of the polymer precursor on the lithium metal.

Supplementary Figure 8. SEM images of the Li/Li symmetrical cell.

Comment 6: PEGDA is a polymer with two C=C end bonds and an ethylene oxide segment ($-CH_2-CH_2-O-$, EO) in the middle. Though SPEs containing EO segments usually exhibit good ionic conductivity, EO segments start to be oxidized at a voltage of over 3.9 V (vs. Li/Li⁺). However, the electrochemical performance of NCM811||Li cells showed no deterioration. Experimental data on the resistance of the SPE to NCM811 oxidation should be provided.

Response: Thank you very much for your comment. In this work, except for PEGDA (containing EO segments), the as-proposed SPE in this work (LASE) also consists of SN and LLZTO@PAN networks, which serve as the plasticizer and reinforcing framework, respectively. The C≡N polar groups belonging to PAN and SN demonstrated high anti-oxidative merits in literature (*Energy Storage Mater.*, 2023, 56, 121-131). Furthermore, the incorporation of LLZTO inorganic components could further enhance the high voltage tolerance of the SPE (*Energy Storage Mater.*, 2022, 51, 130-138). As revealed in the LSV measurement shown in Figure 2e, the LASE exhibited a wide electrochemical stability window of 4.8 V. Based on the above reasons, the LASE showed excellent compatibility with high-voltage cathode NCM811. For better comparison, we also supplemented the LSV curve of PEGDA. As shown in Figure S11, the PEGDA exhibited a much lower electrochemical window of 4.40 V which further illustrated the effect of adding PAN, SN and LLZTO on improving the electrochemical stability window of SPE.

According to your suggestion, we have added the related description in the revised manuscript (Page 8, line 218-220).

Figure 2e Linear sweep voltammetry curves to characterize the electrochemical window of electrolytes.

Supplementary Figure 15. Linear sweep voltammetry curve to characterize the electrochemical window of PEGDA.

Comment 7: Compared with other reported strategies for SPE and inorganic SSE, (for example, *Energy Storage Materials*, 2023, 63, 103009, *Energy Storage Materials*, 2022, 50, 810, *Journal of Colloid and Interface Science*, 2023, 642, 193, and *Nano Energy*, 2022, 91, 106643), what's the advantage and disadvantage of the LAPN-based SE from the aspects of interface property and electrochemical performance?

Response: Thank you very much. We have carefully read the provided literatures. All four papers have done excellent work on improving the Li anode interface property, focusing on directly designing the interface layer to optimize the Li^+ transport in Li anode interface, including constructing Li-Ga alloy and LiCl on the interface of Li/Li₇P₃S₁₁ (*Energy Storage Mater.*, 2023, 63, 103009), construct a Li₂S/Li_xSn mixed ionic/electronic conductive layer (*Nano Energy*, 2022, 91, 106643) and an all-inorganic mixed conductive layer (*Energy Storage Mater.*, 2022, 50, 810) on Li/LLZO, design a TPU artificial SEI layer with good electronic insulation and Li-ion conductivity on Li anode (*J. Colloid Interface Sci.*, 2023, 642, 193). Compared with those literatures, this work improves the lithium anode interface through optimizing the components and anisotropic structure of solid electrolyte (SE) itself. The designed LAPN-based SE with an oriented composite structure is employed to homogenize Li^+ flux, serving as a physical barrier to resist Li dendrites, retarding the side reaction between the electrolyte and Li, thus endowing a compatible interface for Li anodes. Due to the above advantages, the assembled cells using LAPN-based SE show highly improved

electrochemical performance compared with RPN-based SE and APN-based SE. The LFP|LASE|Li cell could steadily operate over 1000 cycles, delivering a high capacity retention of 91% after 1000 cycles at 1C, and the NCM811|LASE|Li cell displays a high capacity retention of 84% after 300 cycles at 0.5 C.

According to your suggestion, we added the related references to the revised manuscript as rf. 7, 8, 9 and 10. The advantages of the LAPN-based SE from interface property and electrochemical performance are also highlighted (Page 4, line 103-105).

“Also, designing a mixed conductive layer in the interface of SSE/Li is demonstrated as an effective strategy to optimize the Li^+ transport in the Li anode interface and then enhance the cycle life of lithium metal batteries.⁷⁻¹⁰”

“Relying on the optimized electrochemo-mechanical interface induced by the as-proposed SPE, the assembled LMBs paired with different cathodes demonstrate outstanding cycling performances and rate capability.”

7. Shi Y, et al. An optimizing hybrid interface architecture for unleashing the potential of sulfide-based all-solid-state battery. *Energy Storage Mater.*, 63, 103009 (2023).
8. Jiang J, et al. In-situ construction of Li-Mg/LiF conductive layer to achieve an intimate lithium-garnet interface for all-solid-state Li metal battery. *Energy Storage Mater.*, 50, 810-818 (2022).
9. Zhao B, et al. A fast and low-cost interface modification method to achieve high-performance garnet-based solid-state lithium metal batteries. *Nano Energy*, 91, 106643 (2022).
10. Zhao B, et al. Construction of high elastic artificial SEI for air-stable and long-life lithium metal anode. *J. Colloid. Interface Sci.*, 642, 193-203, (2023).

Response to Reviewer #2

Original comments:

***Comment 1:** In the Introduction, it would be beneficial to provide the full names for PEGDA, SN, and LiTFSI to aid readers who may not be familiar with these abbreviations. Review other terminologies to ensure they are also clearly specified.*

Response: Thank you very much for your comment. According to your suggestion, we have added the full names when the abbreviations first appear in the revised manuscript.

***Comment 2:** Can the authors provide the rationale for selecting these materials for the LAPN electrolyte, namely PAN, PEGDA, SN, LiTFSI, and LLZTO, instead of other alternatives commonly used in solid polymer electrolytes (SPEs)? This explanation would ideally be placed at the beginning of the Results for clarity of the context.*

Response: Thank you very much for your comment. We apologize for the lack of clarity in explaining the rationale for selecting PAN, PEGDA, SN, LiTFSI, and LLZTO as the core components for the LAPN electrolyte. In formulating this electrolyte, our selection was grounded in the distinctive properties of these materials and their pivotal roles within the electrolyte system, as detailed below:

The principal chain architecture of PAN encompasses a $C\equiv N$ polar group, serving as a Lewis base capable of coordinating with lithium ions and dissociating lithium salts. This $C\equiv N$ polar group possesses a pronounced electron-withdrawing capacity, rendering PAN with a low LUMO and exceptional electrochemical stability. (*J. Am. Chem. Soc.* 2019, 141, 23, 9165–9169; *Energy Storage Mater.*, 2021, 34, 388-416). The PEGDA excels in its cross-linking capabilities and boasts an abundance of polar oxygen-containing groups ($-C-O-C-$, $C=O$) within its chain segments. These groups facilitate effective interactions with Li^+ , thereby significantly bolstering the Li^+ transport capacity of the electrolyte (*Adv. Funct. Mater.* 2023, 33, 2305383). Moreover, the PEGDA monomer is cost-effective, and the in-situ curing technology process is straightforward, making it highly compatible with existing lithium battery

manufacturing processes (*Energy Environ. Mater.* 2023, 6, e12447). The SN (succinonitrile) has a similar main chain structure with PAN, and incorporates C≡N groups that contribute to enhancing the electrochemical window. In addition, SN also contributes to fast ionic transport (*Energy Storage Mater.* 2023, 56, 121-131). LiTFSI, as a lithium salt, is favored for its high ionic conductivity. Moreover, the SEI enriched with LiF, which is formed following the decomposition of TFSI, aids in inhibiting the formation of lithium dendrites, thereby further enhancing the safety of the electrolyte. Finally, the incorporation of LLZTO elevates the electrochemical performance of the electrolyte. Additionally, the interface between PAN and LLZTO may foster intimate chemical interactions, promoting partial dehydrogenation of PAN and yielding localized conjugated structures. These conjugated structures enhance Li⁺ interactions and expedite Li⁺ exchange at the LLZTO/PAN interface, thereby establishing a contiguous Li⁺ conduction pathway and fostering uniform Li deposition. Our theoretical calculations are worth mentioning, which demonstrate that LLZTO further mitigates the reactivity of SN with the Li anode. Beyond electrochemical modulation, the dispersion of LLZTO within PAN fibers contributes to the augmentation of PAN's mechanical properties (*Nano Lett.* 2021, 21, 7070-7078).

According to your suggestion, the related description and relevant references to explain the merits of raw materials, including PAN, PEGDA, SN and LLZTO have been cited in the revised manuscript (Page 4, line 88-97).

“Herein, we propose an anisotropic reinforcement concept by inserting oriented PAN fiber networks into the polymer matrix that was in-situ polymerized by poly(ethylene glycol) diacrylate (PEGDA),²⁴ succinonitrile(SN, as plastic crystals), and lithium bis(trifluoromethanesulphonyl)imide (LiTFSI), aiming to address interfacial drawbacks that derive from the insufficient electrochemical and mechanical properties of SPEs. As revealed in previous studies, both of PAN and SN encompasses C≡N polar group which are demonstrated to contribute to enhance the electrochemical window of electrolyte.^{25,26} Additionally, based on the oriented structure, Li-conductible inorganic fillers (Li_{6.4}La₃Zr_{1.4}Ta_{0.6}O₁₂, LLZTO) are further introduced to ameliorate the functionality of the supporting skeleton.²⁷”

24. Yang M, et al. *Coupling Anion-Capturer with Polymer Chains in Fireproof Gel Polymer Electrolyte Enables Dendrite-Free Sodium Metal Batteries. Adv. Funct. Mater.* 33, 2305383 (2023).
25. Liang J, et al. *Engineering Janus Interfaces of Ceramic Electrolyte via Distinct Functional Polymers for Stable High-Voltage Li-Metal Batteries. J. Am. Chem. Soc.* 141, 9165–9169 (2019).
26. Zhang X, et al. *Novel PEO-based composite electrolyte for low-temperature all-solid-state lithium metal batteries enabled by interfacial cation-assistance. Energy Storage. Mater.* 56, 121-131 (2023).
27. Zhang M, et al. *Flexible, Mechanically Robust, Solid-State Electrolyte Membrane with Conducting Oxide-Enhanced 3D Nanofiber Networks for Lithium Batteries. Nano Lett.,* 21, 7070-7078 (2021).

Comment 3: Distinguishing the difference between the WAXS data of RPN and APN in Figure 1d is challenging. To enhance the visibility of any differences between the two samples, the authors are recommended to plot the intensities of the data.

Response: Thank you very much for your comment. According to your suggestion, we have optimized the visibility of WAXS plots as below.

Figure 1d WAXS patterns of RPN and APN.

Comment 4: The SEM image for LAPN in Figure 1f does not provide conclusive evidence regarding the successful combination of PAN and LLZTO. To address this limitation, the authors are recommended to conduct elemental analysis, such as energy-

dispersive X-ray spectroscopy (EDS), to provide further insight into the composition of the material.

Response: Thank you very much for your comment. According to your suggestion, we have added the EDS mapping of the nanofibers and shown in Figure S2. To further validate the combination of PAN and LLZTO, TEM image of the nanofiber was also conducted. As shown in Figure S3, the LLZTO particle with about 300 nm was embedded in the PAN nanofiber.

According to your suggestion, the EDS mapping of the nanofibers and the corresponding description have been added in the revised manuscript (Page 5, line 136-139).

“As demonstrated in Figure 1f, the protuberance appearing at the fiber surface hinted at the successful combination of PAN fibers and LLZTO particles, which was further validated by the EDS mapping in Supplementary Fig. 2 and the TEM image in Supplementary Fig. 3.”

Supplementary Figure 2. EDS mapping of the nanofibers in LAPN.

Supplementary Figure 3. TEM image of the nanofiber in LAPN.

Comment 5: The 2D diffusion models for RSE, ASE, and LASE in Figure 2b and Supplementary figure 7 should be rigorously demonstrated to accurately reflect their real microstructures and to avoid biased comparisons. Specifically, the total cross-sectional area of PAN fiber should remain constant for each model.

Response: Thank you very much for your comment. According to your suggestion, the domain is modified as follows to enhance comparability between different models:

1. The dimension of diffusion regions remains constant at $250 \times 80 \mu\text{m}$.
2. The number of PAN fiber layers has been increased.
3. The cross-sectional area of PAN fibers in different models is controlled to be a constant value of $6667 \mu\text{m}^2$ (30% of the total area of SPE).

The revised computational domain is illustrated in the figure below (**Supplementary Figure 7**):

Supplementary Figure 7. Schematic demonstration of the 2D diffusion model of RSE, ASE, and LASE.

The numerical simulation results have been recalculated as follows (Figure 2b):

Figure 2b Li⁺ pathway demonstrated by 2D diffusion models.

Comment 6: The XPS results in Fig. 3d and e and their explanations require several revisions: 1) Two peaks at 685.1 (LiF) and 688.4 eV (TFSI⁻) in the F 1s spectra is conversely corresponded as TFSI⁻ and LiF, respectively. 2) Reverse the direction of the x-axis, with lower binding energy to the right. 3) The peak deconvolution appears to have been done somewhat inaccurately. The FWHM and energy values among identical components should match more closely than they currently do. I would recommend redoing the peak deconvolution, as proper interpretation will only be possible thereafter.

Response: Thank you very much for your comment. We apologize for our carelessness about the figures and discussion. You are right that the peaks in the F 1s at 685.1 eV correspond to LiF and 688.9 eV corresponds to -CF₃. According to your suggestion, we revised the XPS to reverse the direction of the x-axis and redo the peak deconvolution. As shown in Figures 3d and 3e. A slightly higher LiF content was detected on the anode surface paired with ASE. In the high-resolution XPS spectrum of N1s (Figure 3e), TFSI⁻/C≡N and Li₃N results from SN and LiTFSI and their decomposed products. With the introduction of LLZTO particles, the Li₃N content decreased from 45.5% to

33.1%, and intriguingly, a new peak was observed at 398.0 eV, representing the appearance of -C=N-C- bonds generated by the side reaction between LLZTO and SN. The decreased Li₃N and the newly formed -C=N-C- suggested that the interaction of LLZTO-SN was propitious to retarding the reaction activity between SN and Li anode. The corresponding description has been added in the revised manuscript (Page 10, line 262-272).

“As shown in Figure 3d, the high-resolution XPS spectra of F1s were deconvoluted into two peaks at 685.1 and 688.9 eV, which corresponded to the -CF₃ component of residual anions (TFSI) and its decomposition products, LiF in SEI layer, respectively.³⁶ In this trial, a little higher LiF content of ≈20.6% was detected on the anode surface paired with ASE. In the high-resolution XPS spectrum of N1s (Figure 3e), the peak at 400.0 eV was assigned to -C≡N, derived from the TFSI/SN.³⁷ The peak at 399.0 eV hinted at the presence of Li₃N, which originated from the reactant products between TFSI/SN and Li anode.³⁷ With the introduction of LLZTO particles, the Li₃N content decreased from 45.5% to 33.1%. Intriguingly, a new peak was observed at 398.0 eV, representing the appearance of -C=N-C- bonds generated by the side reaction between LLZTO and SN.^{38, 39}”

Figure 3. XPS spectra to analyze the chemical composition of the Li anode cycled with ASE and LASE: **d** F1s and **e** N1s.

Comment 7: *The cycle performances of SPEs are evaluated as shown in Figure 4 by pairing them with LFP and NCM cathodes, and Li metal anodes. The results indicate that LASE outperformed ASE; however, with the 20 wt% of LLZTO in LASE compared to ASE, there might be a loss of energy density in the cell as a trade-off for better cyclability. I recommend that the authors to calculate and compare the energy densities (in Wh/kg and Wh/L) of the cells with LASE and ASE.*

Response: Thank you very much for your comment. The weight ratio of LLZTO in LLZTO/PAN nanofiber membrane was 20%. The LASE consisted of polymer electrolytes and LLZTO/PAN nanofibers. By comparing the weight of ASE and LASE, the weight ratio of LLZTO in the LASE was only 4.5%. According to your suggestion, we have calculated the energy densities of the cells with LASE and ASE, the detailed calculations are shown in Table S4. Though the LASE with 20 wt% of LLZTO showed a little higher areal density compared with ASE, the energy density of the LASE-based cell showed higher gravimetric and volume energy density (326.4 Wh kg⁻¹ and 418.2 Wh L⁻¹) than that of ASE-based cell (316.3 Wh kg⁻¹ and 400.2 Wh L⁻¹) due to the low mass proportion of LLZTO in the entire battery (~1.2 wt%) and higher discharge capacity/voltage of LASE-based cell. In the future, by increasing the mass loading of the NCM cathode and reducing the thickness electrolyte membrane, the mass proportion of LLZTO would further decrease, and then the effect of the LLZTO additive in LASE on the energy density of the cell would also be further reduced.

The corresponding description has been added in the revised manuscript (Page 12, line 323-327).

“As revealed in Supplementary Table 4, though the LASE with 20 wt% of LLZTO showed a little higher areal density compared to ASE, the energy density of the LASE-based cell showed higher energy density due to the low mass proportion of LLZTO in the entire battery (~1.2 wt%) and higher discharge capacity/voltage of LASE-based cell.”

Table S4. Calculation of the energy densities of NCM811||Li pouch cells.

Component of pouch cell	Parameter	Value
Cathode (NCM811)	Active material loading (mg cm ⁻²)	9.80
	Cathode loading (mg cm ⁻²)	10.32
	Thickness (μm)	40
ASE	Thickness (μm)	80
	Areal density (mg cm ⁻²)	5.78
LASE	Thickness (μm)	80
	Areal density (mg cm ⁻²)	6.05
Anode (Li)	Thickness (μm)	40
	Areal density (mg cm ⁻²)	2.14
Current collector (Al)	Thickness (μm)	14
	Areal density (mg cm ⁻²)	3.78
Pouch cell NCM811 ASE Li	Total mass (mg cm ⁻²)	22.02
	Total thickness (μm)	174
	Areal capacity (mAh cm ⁻²)	1.86
	Average discharge voltage (V)	3.744
	Gravimetric energy density (Wh kg ⁻¹)	316.3
Pouch cell NCM811 LASE Li	Volume energy density (Wh L ⁻¹)	400.2
	Total mass (mg cm ⁻²)	22.29
	Total thickness (μm)	174
	Areal capacity (mAh cm ⁻²)	1.94
	Average discharge voltage (V)	3.751
Pouch cell NCM811 LASE Li	Gravimetric energy density (Wh kg ⁻¹)	326.4
	Volume energy density (Wh L ⁻¹)	418.2

The end.

Title: An Anisotropic Strategy for Developing Polymer Electrolytes Endowing Lithium Metal Batteries with Electrochemo-mechanically Stable Interface

Authors: Jingren Gou, Kaixun Cui, Zheng Zhang, Suqing Wang, Jiale Huang, Haihui Wang

Many thanks to the reviewers for their valuable comments and suggestions. The followings are the point-by-point response to the concerns:

Response to Reviewer #1

Original comments:

In my opinion, the manuscript is well-organized and the authors have given sufficient answers to the Reviewers' comments. Therefore, the manuscript can be published in this journal.

Response: Thank you very much for your encouragement.

Response to Reviewer #2

Original comments:

The authors have thoroughly addressed all comments, and the changes made have improved the quality of the manuscript. Their responses to some of my comments have been satisfactory, however, not all issues from the first review have been resolved.

The authors added more analysis as I suggested, but the evidence for the results remains arguable, making the discussions controversial. The EDS and TEM images in Supplementary Figure 2 and 3 do not provide further evidence. It is recommended to conduct a more robust experiment to identify the successful combination of PAN and LLZTO. The XPS data in Figures 3d and e were displayed in the correct format, but the deconvoluted peaks and their explanations are still unclear; further experiments are required to find the role of LLZTO.

Overall, the authors provide a comprehensive explanation of the aligned PAN, but they

could not adequately elucidate on the reasonings regarding LLZTO. A more robust analysis or stronger evidence is required to improve the reliability of this work. Currently, the manuscript seems insufficient for publication.

Response: Thank you very much for your time and valuable suggestions. We have revised our manuscript according to your professional suggestions, and we hope the supplement characterization and explanations can help readers better understand our work. We have made point-by-point responses to your comments as follows.

According to your suggestion, we have retested the TEM images supplemented by the HAADF-STEM image and corresponding EDS mappings of the PAN/LLZTO nanofibers to identify the successful combination of PAN and LLZTO particles. As shown in Figure R1 and Figure R2, the LLZTO particles with a diameter of ~ 200 nm were anchored in the PAN nanofibers. The EDS mappings reveal a homogeneous distribution of the constituent elements throughout the material. Specifically, C and N are uniformly dispersed within the PAN nanofibers, while La, Zr, Ta and O are predominantly localized within the LLZTO ceramic particles.

Figure R1. The TEM images of the nanofiber in LAPN.

Figure R2. The EDS mapping of the nanofibers in LAPN.

According to your comments, we have added Figures R1 and R2 as Supplementary Fig. 2 and Supplementary Fig. 3 in the revised supporting information. The corresponding description has been added in the revised manuscript (Page 5, line136-142).

“As depicted in Figure 1f and Supplementary Fig. 2, the LLZTO particles, approximately 200 nm in diameter, are interspersed within the PAN nanofiber matrix, demonstrating the successful integration of PAN and LLZTO components. In addition, EDS mapping (Supplementary Fig. 3) reveals a homogeneous distribution of the constituent elements throughout the material. Specifically, C and N are uniformly dispersed within the PAN nanofibers, while La, Zr, Ta and O are predominantly localized within the LLZTO ceramic particles.”

To explain the role of LLZTO in the interface layer between Li anode and electrolyte, we have supplemented the different depth XPS tests (Figure R3) and the time-of-flight secondary ion mass spectrometry (ToF-SIMS, Figure R4) of the cycled Li anodes.

As shown in Figure R3a-b, the high-resolution XPS spectra of the F1s region are deconvoluted into two peaks at 688.9 and 685.3 eV, corresponding to the -CF₃ and LiF components in SEI layer, respectively. With increasing sputtering time, the content of LiF gradually increases. The content of LiF in the LASE electrolyte is consistently higher than that in the ASE. In the high-resolution XPS spectra of the N1s region (Figure R3c-d), the peak at 400.0 eV is attributed to the -C≡N group, resulting from the reaction between TFSI-/SN and the Li anode.³⁸ The peak at 399.0 eV suggests the presence of Li₃N. Interestingly, a new peak at 398.0 eV, representing the -C=N-C- group, is observed, indicating a reaction between LLZTO and SN.

To further analyze the spatial distribution of the SEI, the TOF-SIMS tests of the cycled Li anodes were employed. As shown in Figure R4, throughout the entire sputtering duration, the intensity of the inorganic component signals from the Li anode paired with the LASE electrolyte was significantly higher than that from the Li anode paired with the ASE. Moreover, the corresponding 3D profiles of the fragments indicated the formation of a uniform SEI on the Li anode, which can facilitate Li⁺ transport and suppress the growth of Li dendrites.

Figure R3. XPS spectra to analyze the chemical composition of the Li anode cycled with ASE and LASE: (a,b) F1s and (c,d) N1s.

Figure R4. (a) ToF-SIMS depth profiles of various atom group fragments formed on the Li anode after cycling. (b) ToF-SIMS 3D profiles of LiF_2^- and Li_3N^- for the Li anode cycled with ASE and LASE.

According to your comments, we have added Figure R3 and Figure R4 as Supplementary Fig.17, Figure 3d and Supplementary Fig. 20 in the revised manuscript, as well as revised supporting information. The corresponding description has been added in the revised manuscript (Page 10, line 264-278; Page 11, line 281-289).

“As illustrated in Supplementary Fig. 17, the high-resolution XPS spectra of the F1s region are deconvoluted into two peaks at 688.9 and 685.3 eV, corresponding to the -CF₃ and LiF components in SEI layer, respectively.³⁶ With increasing sputtering time, the content of LiF gradually increases. The content of LiF in the LASE electrolyte is consistently higher than in the ASE, which helps to stabilize the interface.³⁷ In the high-resolution XPS spectra of the N1s region, the peak at 400.0 eV is attributed to the -C≡N group, resulting from the reaction between TFSI/SN and the Li anode.³⁸ The peak at 399.0 eV suggests the presence of Li₃N. Interestingly, a new peak at 398.0 eV, representing the -C=N-C- group, is observed, indicating a reaction between LLZTO and SN.³⁹ As the sputtering time increases, the content of Li₃N gradually increases while the content of -C=N-C- remains relatively stable. Li₃N can act as a fast Li conductor, enhancing the stability and electrical conductivity of the SEI, reducing interfacial impedance, and improving cyclic stability.⁴⁰”

“The time-of-flight secondary ion mass spectrometry (ToF-SIMS) was further employed to analyze the spatial distribution of the SEI. The TOF-SIMS detected atomic fragments related to the inorganic components LiF₂⁻ and Li₃N⁻ (Figure 3d and Supplementary Fig. 20). Notably, throughout the entire sputtering duration, the intensity of the inorganic component signals from the Li anode with the LASE electrolyte was significantly higher than that from the Li anode with the ASE. Moreover, the corresponding 3D profiles of the fragments indicated the formation of a uniform SEI on the Li anode, which can facilitate Li⁺ transport and suppress the growth of Li dendrites.^{41,42}”

Additional experiments have been conducted and are presented herein to elucidate the role of LLZTO further.

Initially, the chemical compatibility between LLZTO and SN was examined using XRD. As depicted in Figure R5a, the characteristic peaks remained unchanged after mixing

LLZTO with SN, indicating good chemical compatibility. Subsequently, FTIR was conducted on SN and SN+LLZTO to explore the interaction between LLZTO and SN (Figure R5b). The vibrational peak observed at 2252.9 cm^{-1} corresponds to the $\text{C}\equiv\text{N}$ stretching mode of SN, and the decrease in peak intensity upon the addition of LLZTO may be due to the interaction between the $\text{C}\equiv\text{N}$ groups of SN and LLZTO. Further analysis of the electronic aggregation state of LLZTO in the SN+LLZTO mixture was performed using XPS, and the results are shown in Figures R5c,d. The positive shift observed in the La 3d spectra after the addition of SN indicates an increase in the binding energy of La 3d. This may be attributed to the interaction between La atoms (in LLZTO) and N atoms (in SN), which reduces in the electronic density around the La atoms.

Figure R5. XRD patterns of SN, LLZTO and SN+LLZTO. (b) FTIR spectra of SN and SN +LLZTO. XPS spectra of (c) LLZTO and (d) SN +LLZTO.

The addition of LLZTO could enhance the interface stability of electrolyte with Li anode due to the interaction between La^{3+} in LLZTO and N atoms in SN can reduce the activity of CN triple bond groups or convert them into weaker CN double bond groups, avoiding SN corrosion of Li anode. To better support this point, we supplemented the LSV curves of the cells using ASE and LASE at a scan rate of $1\text{ mV} \cdot \text{s}^{-1}$ at the potential range from open circuit voltage to -0.2 V . As shown in Figure R6, the ASE electrolyte

exhibits reduction around 1.0 V. In contrast, the LASE electrolyte demonstrates superior resistance to reduction, indicating that the LLZTO in LASE can enhance the interfacial stability with the Li anode.

Figure R6. LSV curves of the cells using ASE and LASE at a scan rate of $1 \text{ mV} \cdot \text{s}^{-1}$ in the potential range from open circuit voltage to -0.2 V.

According to your comments, we have added Figure R5 and Figure R6 as Supplementary Fig. 18 and Supplementary Fig. 19 in the revised supporting information. The corresponding description has been added in the revised manuscript (Page 10-11, line 277-281) and revised supporting information (Page S9-S10).

“Further analytical characterizations confirmed the interaction between LLZTO and SN (Supplementary Fig. 18-19). The interaction between La^{3+} in LLZTO and N atoms in SN can reduce the activity of $-\text{C}\equiv\text{N}$ groups or convert them into weaker $-\text{C}=\text{N}-\text{C}$ group groups, preventing the corrosion of the Li by SN molecules and enhancing the interface stability of electrolyte with Li anode.³⁸”

Based on the provided characterizations, the beneficial contributions of LLZTO in the LASE could be summarized as follows: 1. Enhance the mechanical properties of the electrolyte; 2. Lower the activation energy (E_a) and improve the ionic conductivity of the electrolyte. 3. Wider the electrochemical stable window of the electrolyte, which helps to pair with high-voltage cathodes; 4. Improve the interface stability with Li anode and then extend the lifespan of the batteries.

The end.